# p21-activated kinase 1 restricts tonic endocannabinoid signaling in the hippocampus

Shuting Xia[1†], Zikai Zhou[1,2†], Celeste Leung[3,4], Yuehua Zhu[1], Xingxiu Pan[1], Junxia Qi[1], Maria Morena[5,6], Matthew N Hill[5,6*], Wei Xie[1,2*], Zhengping Jia[3,4*]

[1]The Key Laboratory of Developmental Genes and Human Disease, Jiangsu Co-innovation Center of Neuroregeneration, Southeast University, Nanjing, China; [2]Institute of Life Sciences, Collaborative Innovation Center for Brain Science, Southeast University, Nanjing, China; [3]Neurosciences and Mental Health, The Hospital for Sick Children, Toronto, Canada; [4]Department of Physiology, Faculty of Medicine, University of Toronto, Toronto, Canada; [5]Hotchkiss Brain Institute, Cumming School of Medicine, Calgary, Canada; [6]Department of Cell Biology and Anatomy and Psychiatry, University of Calgary, Calgary, Canada

**Abstract** PAK1 inhibitors are known to markedly improve social and cognitive function in several animal models of brain disorders, including autism, but the underlying mechanisms remain elusive. We show here that disruption of PAK1 in mice suppresses inhibitory neurotransmission through an increase in tonic, but not phasic, secretion of endocannabinoids (eCB). Consistently, we found elevated levels of anandamide (AEA), but not 2-arachidonoylglycerol (2-AG) following PAK1 disruption. This increased tonic AEA signaling is mediated by reduced cyclooxygenase-2 (COX-2), and COX-2 inhibitors recapitulate the effect of PAK1 deletion on GABAergic transmission in a CB1 receptor-dependent manner. These results establish a novel signaling process whereby PAK1 upregulates COX-2, reduces AEA and restricts tonic eCB-mediated processes. Because PAK1 and eCB are both critically involved in many other organ systems in addition to the brain, our findings may provide a unified mechanism by which PAK1 regulates these systems and their dysfunctions including cancers, inflammations and allergies.

*For correspondence: mnhill@ucalgary.ca (MNH); wei.xie@seu.edu.cn (WX); zhengping.jia@sickkids.ca (ZJ)

†These authors contributed equally to this work

## Introduction

It is generally accepted that normal brain function is dependent upon a balance of excitation and inhibition (i.e. balanced E/I ratio) and that altered E/I ratios are associated with, and thought to cause, a wide range of neurological and mental disorders, including autism and schizophrenia (*Eichler and Meier, 2008*; *Kehrer et al., 2008*; *Marín, 2012*; *Yizhar et al., 2011*). Recent studies indicate that GABA-mediated synaptic inhibition is particularly important to maintain an appropriate E/I ratio and that inhibitory postsynaptic currents (IPSCs) mediated by GABA receptors are frequently altered in various brain diseases and their animal models (*Braat and Kooy, 2015*; *Lewis et al., 2005*). Although subjected to multiple regulations, GABA transmission is strongly inhibited by endocannabinoids (eCBs), a group of neuromodulatory lipids known to affect a wide range of physiological processes and medical conditions (*Katona and Freund, 2012*; *Morena et al., 2016*; *Piomelli, 2003*). In the brain, eCBs are produced and secreted from postsynaptic neurons and activate presynaptic cannabinoid 1 (CB1) receptors to reduce the release of a multitude of neurotransmitters, including GABA (*Katona and Freund, 2012*; *Morena et al., 2016*; *Piomelli, 2003*). Two types of eCB-mediated suppression of GABA release have been studied: tonic eCB release that

**eLife digest** Brain cells communicate by sending chemical signals that activate or excite neighbouring cells. However, too much signalling can be harmful. As such the brain has systems in place to inhibit brain signals, and healthy brain activity relies striking a proper balance between excitation and inhibition. In some brain mental health conditions, like autism or schizophrenia, the balance is skewed which has an impact on the brain's activity.

A chemical produced by brain cells called endocannabinoid helps maintain the appropriate balance in brain excitation and inhibition. Endocannabinoid is similar to a chemical found in cannabis, but little is known about how it works and which proteins interact with endocannabinoid. A family of proteins called p21-activated kinases (PAKs) has been implicated in autism and other disorders like Huntingtin disease and Alzheimer disease, but it is not fully understood how these proteins interact with endocannabinoid.

Now, Xia, Zhou et al. show that one member of this protein family called PAK1 plays a key role in controlling endocannabinoid activity. The experiments showed that mice genetically engineered to lack the PAK1 protein have higher levels of endocannabinoids and, as a consequence, the chemical signals that inhibit brain cells are affected more. The experiments also revealed that PAK1 does not interact directly with endocannabinoids. Instead PAK1 boosts levels of another protein called COX-2 and reduces levels of a molecule called anandamide, which together restrict endocannabinoid's inhibitory effects.

Scientists are currently interested in developing drugs that target the endocannabinoids and their regulators in the brain as a way to treat anxiety, pain and sleep problems. Drugs that block PAK1 are already being studied. Future studies are needed to determine if such PAK1-targeting drugs could be useful for restoring excitatory and inhibitory balance in brain diseases or for treating other diseases involving the PAK proteins.

regulates basal synaptic transmission, and phasic eCB release, induced by postsynaptic depolarization or receptor-mediated eCB production, which mediates transient decreases in synaptic transmission during short-term plasticity (*Katona and Freund, 2012*; *Morena et al., 2016*). Although the metabolic process of the eCBs and the enzymes involved in their regulation have been a focus of extensive research (*Katona and Freund, 2012*; *Morena et al., 2016*; *Piomelli, 2003*), cellular signaling mechanisms that regulate eCB signaling, particularly tonic eCB signaling, remain poorly understood.

p21-activated kinases (PAKs) are a family of serine/threonine protein kinases that are activated by multiple synaptic proteins including Ras and Rho GTPases. Extensive studies have indicated that PAKs are involved in a number of cellular processes, particularly in the regulation of gene expression and cellular cytoskeleton (*Bokoch, 2003*; *Zhao and Manser, 2012*). Accordingly, changes in PAKs are found to be associated with a wide range of physiological and pathological conditions including various forms of cancers and PAK inhibitors are being actively exploited as therapeutic agents to treat these diseases (*Kelly and Chernoff, 2012*; *Kumar et al., 2006*). Recent human studies have also revealed that PAKs are linked to a number of devastating neurological and mental disorders including autism, intellectual disability, Huntingtin's diseases and Alzheimer's diseases (*Gilman et al., 2011*; *Ma et al., 2012*). Animal studies have indeed shown that PAKs, particularly PAK1, the predominant member of the PAK family expressed in the brain, are involved in the regulation of excitatory synaptic function, including spine structure, synaptic plasticity and memory formation (*Asrar et al., 2009*; *Hayashi et al., 2004*; *Huang et al., 2011*; *Meng et al., 2005*). Most remarkably, more recent studies demonstrate that inhibition of PAK1, either genetically or pharmacologically, can ameliorate the cognitive and social deficits in several animal models of neurodevelopmental disorders, particularly autism, including genetic models targeting fragile X syndrome and neurofibromatosis (*Dolan et al., 2013*; *Hayashi et al., 2007*; *Molosh et al., 2014*). However, the mechanism by which PAK1 exerts such diverse therapeutic effects remains elusive. Quite interestingly, many of these same animal models of neurodevelopmental disorders also exhibit pronounced alterations in eCB signaling (*Földy et al., 2013*; *Jung et al., 2012*), and several reports have now

suggested that, like PAK1, targeting eCB signaling may provide benefit in these conditions (*Busquets-Garcia et al., 2013*; *Qin et al., 2015*). As these animal models share a common deficit in E/I balance, which appear to involve critical roles of both PAK1 and eCB signaling, we have hypothesized that PAK1 might be a critical player in the regulation of E/I homeostasis through an interaction with eCB signaling. Consistent with this hypothesis, our data indicate that PAK1 restricts tonic eCB signaling in the hippocampus through the regulation of synaptosomal cyclooxygenase-2 (COX-2) expression, a non-canonical but relevant pathway in the metabolism of eCB signaling. In turn, this ability of PAK1 to restrict tonic eCB signaling confers an alteration in the E/I homeostasis of the hippocampus through the regulation of tonic GABA transmission. Given the overlapping importance of PAK1, COX-2 and eCB signaling in an array of physiological and pathophysiological processes, the identification of this functional signaling interaction likely has significant implications for a multitude of disease processes, such as autism, inflammatory conditions and cancer.

## Results

### PAK1 disruption enhances the E/I ratio by suppressing inhibitory synaptic transmission

Since all the animal models of brain disorders that are functionally rescued by manipulations of PAK1 share a common deficit in E/I balance (*Braat and Kooy, 2015*; *Dolan et al., 2013*; *Eichler and Meier, 2008*; *Gao and Penzes, 2015*; *Hayashi et al., 2007*; *Kehrer et al., 2008*; *Lewis et al., 2005*; *Marín, 2012*; *Molosh et al., 2014*; *Yizhar et al., 2011*), we therefore examined whether disrupting PAK1 would affect the E/I ratio by performing whole cell patch-clamp recordings in CA1 pyramidal neurons of hippocampal slices acutely prepared from PAK1 KO mice and their WT littermates (*Figure 1a*). Excitatory and inhibitory postsynaptic currents (EPSCs and IPSCs) were pharmacologically isolated by using respective inhibitors specific to glutamate or GABA receptors (i.e. EPSC recorded by including 100 μM GABAα receptor antagonist picrotoxin and IPSC by including 10 μM AMPAR antagonist NBQX plus 50 μM NMDAR antagonist D-APV). First, we measured the E/I ratio by sequentially recording evoked synaptic responses (*Figure 1b*), first in the absence of any inhibitors to obtain total synaptic currents (i.e. eEPSC+eIPSC), then in the presence of NBQX/APV to obtain eIPSC, and finally in the presence of NBQX/APV/picrotoxin to verify the eIPSC component. As shown in *Figure 1c*, the E/I ratio was significantly increased in PAK1 KO compared to the WT littermates. Because PAK1 KO mice show no deficits in basal excitatory synaptic strength (*Asrar et al., 2009*), the increased E/I ratio in the KO mice is likely due to impaired inhibitory transmission. To test this possibility directly, we performed input/output experiments of eIPSC and as shown in *Figure 1d*, the amplitude of eIPSC was significantly smaller in the KO compared to the WT control over a wide range of stimulus intensities. To exclude the possibility that the KO mice may have suffered developmental compensations that could contribute to the reduced eIPSC, we tested the effect of the group1 PAK inhibitor IPA3 (10 μM) (*Rudolph et al., 2013*). As shown in *Figure 1e*, bath application of IPA3 caused a rapid and significant decrease in the amplitude of eIPSC in WT, but not in PAK1 KO neurons. To further corroborate this result, we employed an independent short peptide known to specifically inhibit PAK1 (*Shin et al., 2013*). As shown in *Figure 1f*, inclusion of this peptide in the postsynaptic neurons (20 μg/ml) also significantly decreased the amplitude of eIPSC in WT, but not in PAK1 KO neurons. The fast acting nature of these inhibitors (within minutes) indicate that the effect of PAK1 on eIPSC is not likely due to a developmental effect but rather direct involvement of PAK1 at the synapse. These results also indicate that PAK1 disruption specifically in the postsynaptic CA1 neurons is sufficient to cause impaired inhibitory synaptic transmission.

### PAK1 deletion specifically reduces the frequency, but not the amplitude of inhibitory synaptic responses

To investigate whether the reduced inhibitory transmission is caused by pre- and/or postsynaptic changes, we recorded spontaneous IPSC (sIPSC) and miniature IPSC (mIPSC). As shown in *Figure 2a–f*, the frequency of sIPSC (*Figure 2c*) and mIPSC (*Figure 2f*) was significantly reduced in PAK1 KO compared to WT neurons. The amplitude of sIPSC (*Figure 2b*) and mIPSC (*Figure 2e*) were not altered in the KO mice. Consistent with previous results (*Asrar et al., 2009*), neither the frequency nor the amplitude of spontaneous or miniature excitatory synaptic currents (sEPSC or

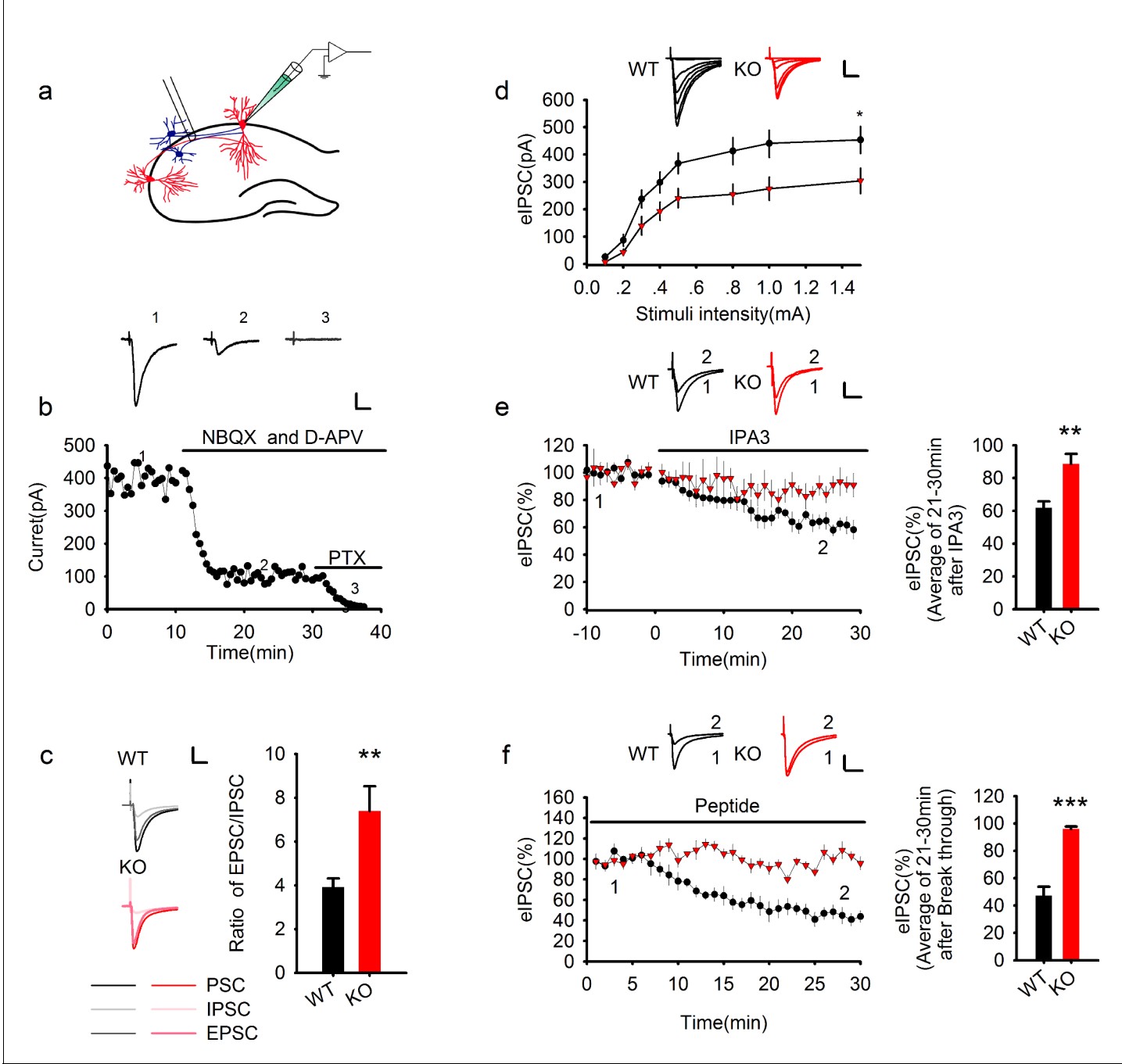

**Figure 1.** Genetic ablation of PAK1 enhances E/I ratio by selectively suppressing inhibitory synaptic responses. (**a**) Diagram of a hippocampal slice showing the placement of stimulating and recoding electrodes. (**b**) A representative whole-cell recording experiment and samples traces at indicated time points showing the time course of evoked synaptic currents in the absence or presence of various inhibitors to determine the E/I ratio. Scale bar: 100 pA/25 ms. (**c**) Left: sample traces of various components of synaptic currents. Right: summary data showing an increased E/I ratio in PAK1 KO compared to WT control (WT = 3.92 ± 0.39, n = 8 (5); KO = 7.39 ± 1.13, n = 6 (4); **p=0.007; t-test). Scale bar: 100 pA/25 ms. (**d**) Whole-cell recordings of input-output curves showing significantly reduced amplitude of evoked IPSC (eIPSC) in PAK1 KO compared to WT neurons (genotype: F(1, 27) = 5.946, *p=0.022; stimuli: F(7, 189) = 70.983, ***p<0.001; repeated measures two-way ANOVA [also see *Figure 1—source data 1*]; at 1.5 mA stimulus: WT = 453.39 ± 49.41 pA, n = 15 (5); KO = 304.08 ± 46.54 pA, n = 14 (5); *p=0.037; t-test). Scale bar: 125 pA/25 ms. (**e**) Whole-cell recordings of eIPSC showing that bath application of IPA3 caused a rapid decrease in eIPSC amplitude in WT, but not in PAK1 KO neuron (genotype: F(1, 10) = 5.615, *p=0.039; time: F(3, 30) =16.332, ***p<0.001; repeated measures two-way ANOVA [also see *Figure 1—source data 2*]; at 21–30 min post IPA3 perfusion: WT = 61.84 ± 4.01%, n = 7 (4); KO = 88.65 ± 6.08%, n = 5 (3); **p=0.003; t-test). Scale bar: 20 pA/25 ms. (**f**) Whole-cell recordings of eIPSC showing that intracellular infusion of the PAK1 inhibitory peptide specifically in the postsynaptic neurons caused a rapid decrease in eIPSC amplitude in

*Figure 1 continued on next page*

*Figure 1 continued*

WT, but not in PAK1 KO neurons (genotype: F(1, 9) = 62.66, ***p<0.001; time: F(2, 18) = 30.720, ***p<0.001; repeated measures two-way ANOVA [also see *Figure 1—source data 3*]; at 21–30 min after whole-cell break-in: WT = 47.07 ± 6.63%, n = 5 (4); KO = 96.17 ± 1.67%, n = 6(5);***p<0.001; t-test). Scale bar: 40 pA/25 ms.

The following source data is available for figure 1:

**Source data 1.** Statistical data summary for *Figure 1d*: input/output curves of eIPSC using repeated measures two-way ANOVA.
**Source data 2.** Statistical data summary for *Figure 1e*: IPA3 effect on eIPSC using repeated measures two-way ANOVA.
**Source data 3.** Statistical data summary for *Figure 1f*: PAK1 inhibitory peptide effect on eIPSC using repeated measures two-way ANOVA.

mEPSC) was altered in PAK1 KO mice (*Figure 2g–i*). To test whether these changes were specific to PAK1 KO mice, we analyzed KO mice lacking ROCK2 (*Zhou et al., 2009*), a closely related kinase also activated by the Rho family small GTPases, but found no significant changes in any of these parameters in these mice (*Figure 2—figure supplement 1*). These results suggest that the release property at the inhibitory synapse is selectively impaired in PAK1 KO neurons. To investigate this further, we examined transmitter release induced by sustained high frequency stimulations. As shown in *Figure 2—figure supplement 2*, synaptic depression induced by 3 min of 5 Hz stimulations was significantly slower in PAK1 KO mice compared to the WT littermates. These results indicate that PAK1 regulates inhibitory synaptic transmission likely through a presynaptic mechanism.

## Acute PAK1 inhibition in postsynaptic neurons also specifically reduces the frequency, but not the amplitude of inhibitory synaptic responses

Similarly, to exclude the possibility of developmental compensations in the KO animals, we also tested the effect of IPA3 (10 µM) on sIPSC and mIPSC. Again the drug was included in the recording electrode to specifically inhibit PAK1 in the postsynaptic neurons. As shown in *Figure 3a–c*, inclusion of IPA3 caused a significant decrease in the frequency (*Figure 3c*), but not the amplitude (*Figure 3b*) of sIPSC in WT neurons. IPA3 also significantly reduced the frequency (*Figure 3d,f*), but not the amplitude (*Figure 3d,e*) of mIPSC in WT neurons. The frequency of sIPSC and mIPSC in IPA3 treated WT neurons was similar to that of PAK1 KO neurons (*Figure 2c,f*; *Figure 3c*). Importantly, IPA3 had no effect on the frequency of sIPSC and mIPSC in PAK1 KO mice (*Figure 3a,c,d,f*), again confirming that the effect of IPA3 is mediated by PAK1. Neither the frequency nor the amplitude of sEPSC or mEPSC was affected by IPA3 (*Figure 3g–i*). The rapid and selective effect of IPA3 on the frequency of sIPSC and mIPSC again suggests that disruption of postsynaptic PAK1 is sufficient to reduce presynaptic release of inhibitory neurotransmitters.

## The effect of PAK1 is independent of postsynaptic GABA receptors or the actin cytoskeleton

Although the above results that postsynaptic inhibition of PAK1 causes a reduction in the frequency, but not the amplitude of IPSCs, suggest a presynaptic mechanism, it is possible that a reduction in postsynaptic GABA receptors, which could result in silent synapses, may also lead to a reduced frequency in sIPSC and mIPSC. To address this possibility, we first examined the number of GABA-positive neurons and inhibitory synapses, and the level of GABA receptor associated proteins, but found no significant differences between WT and PAK1 KO mice (*Figure 4a–e*; also see *Figure 4—figure supplement 1*). To determine if GABA receptors were functionally equivalent, we recorded IPSCs evoked by a brief puff of GABA (1 mM, 100 ms) in a co-culture system where WT and PAK1 KO neurons were grown on the same coverlips in order to minimize the differences in culturing conditions between genotypes (see Materials and methods), but again found no differences between WT and PAK1 KO neurons (*Figure 4f,g*). To further examine if other postsynaptic processes could contribute to the effect of PAK1, we analyzed the effect of cytochalasin D (5 µM) and NSC23766 (250 µM), pharmacological inhibitors for actin polymerization and Rac1 activation respectively. Both actin and Rac1 are key targets of PAK1 signaling (*Bokoch, 2003*; *Zhao and Manser, 2012*). As shown in *Figure 5a–f*, no effect on sIPSCs was observed for either inhibitor. It is important to note that

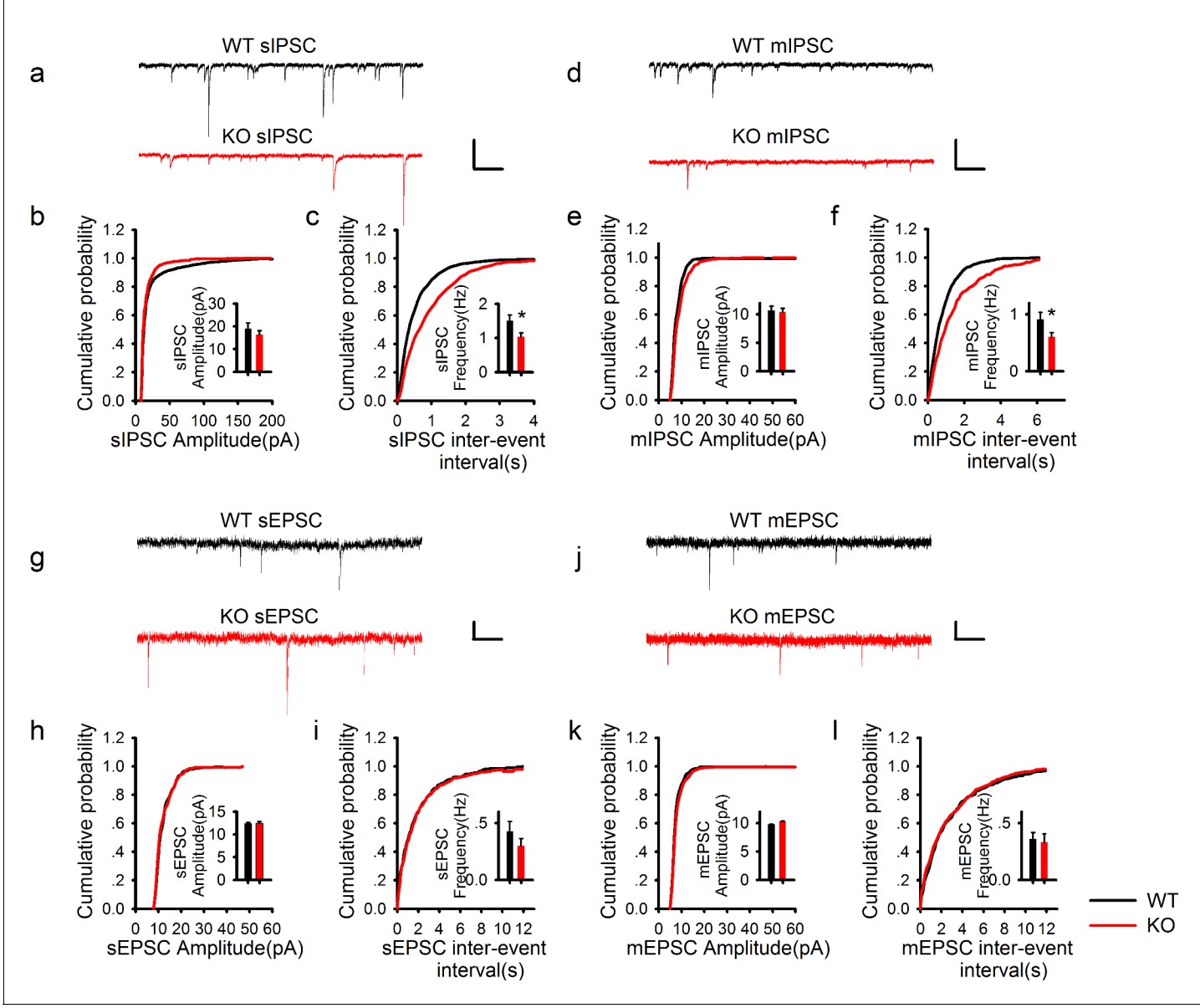

**Figure 2.** PAK1 deletion specifically reduces the frequency, but not the amplitude of inhibitory synaptic responses. (**a**) Sample traces of sIPSC recordings. (**b**, **c**) Summary graphs of (**a**) showing normal distribution and mean value of the amplitude (**b**: WT = 19.07 ± 2.44 pA, n = 13 (5); KO = 16.44 ± 1.62 pA, n = 20 (6); p=0.35), but decreased frequency (**c**: WT = 1.50 ± 0.17 Hz, n = 13 (7); KO 1.03 ± 0.11 Hz, n = 20 (6); *p=0.021) of sIPSCs in PAK1 KO compared to WT control. (**d**) Sample traces of mIPSC recordings. (**e**, **f**) Summary graphs of (**d**) showing normal distribution and mean value of the amplitudes (**e**: WT = 10.69 ± 0.75 pA, n = 10 (4); KO = 10.40 ± 0.64 pA, n = 16 (4); p=0.773), but decreased frequency (**f**: WT = 0.91 ± 0.12 Hz, n = 10 (4); KO = 0.60 ± 0.08 Hz, n = 16 (4); *p=0.034) of mIPSCs in PAK1 KO compared to WT control. (**g**) Sample traces of sEPSC recordings. (**h**, **i**) Summary graphs of (**g**) showing normal amplitude (**h**: WT = 12.37 ± 0.30 pA, n = 20 (5); KO = 12.43 ± 0.46 pA, n = 14 (5); p=0.897) and frequency (**i**: WT = 0.43 ± 0.08 Hz, n = 20 (5); KO = 0.30 ± 0.06 Hz, n = 14 (5); p=0.272) of sEPSCs in PAK1 KO compared to WT control. (**j**) Sample traces of mEPSC recordings. (**k**, **l**) Summary graphs of (**j**) showing normal amplitude (**k**: WT = 9.79 ± 0.06 pA, n = 7 (3); KO = 10.30 ± 0.06 pA, n = 9 (3); p=0.545) and frequency (**l**: WT = 0.36 ± 0.05 Hz, n = 7 (3); KO = 0.33 ± 0.07 Hz, n = 9 (3); p=0.806) of mEPSCs in PAK1 KO compared to WT control. All scale bars: 50 pA/1 s.

The following source data and figure supplements are available for figure 2:

**Figure supplement 1.** Normal inhibitory transmission in ROCK2 KO mice.

**Figure supplement 1—source data 1.** Statistical data summary for *Figure 2—figure supplement 1*: Normal inhibitory transmission in ROCK2 KO mice using one-way ANOVA.

*Figure 2 continued on next page*

*Figure 2 continued*

**Figure supplement 2.** Impaired transmitter depletion in response to sustained synaptic activation.

consistent with previous results (*Meng et al., 2002*; *Zhou et al., 2011*), these two inhibitors (at the same concentrations used here) had profound effects on excitatory synaptic function, including basal synaptic transmission (*Figure 5g–i*) and metabotropic glutamate receptor (mGluR)-dependent long-term depression induced by 50 µM DHPG (*Figure 5j,k*). Taken together, we concluded that postsynaptic PAK1 regulates inhibitory synaptic transmission likely through a retrograde mechanism to modulate GABA release.

## Endocannabinoid system is enhanced by PAK1 disruption

Endocannabinoids (eCB) are known to be generated and secreted from postsynaptic pyramidal neurons to act as a retrograde messenger to inhibit GABA release (*Katona and Freund, 2012*; *Piomelli, 2003*). While the tonic secretion of eCB affects basal synaptic transmission, its phasic secretion induced by postsynaptic depolarization regulates synaptic plasticity (*Katona and Freund, 2012*; *Morena et al., 2016*; *Piomelli, 2003*). An enhanced tonic signaling would reduce the probability of GABA release, and thus decrease IPSC frequency similar to what we observed in neurons of PAK1 KO mice or in WT neurons loaded with PAK1 inhibitors. Thus, we hypothesized that disruption of PAK1 would enhance tonic eCB signaling. To test this hypothesis, we first examined the effect of AM251, a CB1 receptor antagonist and inverse agonist. In WT, bath application of AM251 (5 µM) increased eIPSCs to approximately 150% of the baseline response (*Figure 6a*), reflecting disinhibition of GABA release by blocking tonically active CB1 receptors (*Neu et al., 2007*). Remarkably, in PAK1 KO mice, AM251 enhanced eIPSC amplitudes to 250% of the baseline response (*Figure 6a*). Acute inhibition of postsynaptic PAK1 by IPA3 (10 µM) or by the PAK1 inhibitory peptide (20 µg/ml) prior to the AM251 application produced similar results as shown in PAK1 KO mice (*Figure 6b,c*). These findings indicate that tonic eCB signaling is restricted by PAK1 and that disruption of PAK1 causes a robust increase in tonic eCB effect. Consistent with this, an increase in tonic eCB signaling could explain why the frequency of sIPSC/mIPSC is reduced in PAK1 KO mice as shown in *Figure 2* and *3*. To further investigate whether the enhanced eCB signaling in the KO mice is due to changes in CB receptors and/or their downstream signaling processes in the presynaptic terminal, we examined the protein level of CB1 receptors, but found no differences in either total brain lysates (*Figure 6d*) or synaptosomal fractions (*Figure 6e*). In addition, bath application of the CB1 receptor agonist WIN (5 µM), designed to maximally activate the receptor, depressed synaptic responses to the same degree in both WT and PAK1 KO mice (*Figure 6f*), suggesting that CB1 receptors and their downstream events are intact in the KO mice. It is important to note that DMSO (the dissolvent for many of the pharmacological agents in this study) had no effect on eIPSC (*Figure 6—figure supplement 1*).

## Elevated AEA, reduced COX-2, but intact 2-AG in PAK1 KO mice

The increased effect of AM251 in the absence of any changes in the level of CB1 receptors or their activation strength suggests that the levels of the eCB molecules might be elevated by PAK1 disruption. To test this possibility directly, we measured the tissue level of AEA and 2-AG in the hippocampus. Interestingly, tissue levels of AEA (*Figure 7a*), but not 2-AG (*Figure 7b*), were significantly elevated in PAK1 KO mice. These data are consistent with the belief that AEA mediates tonic actions of the eCB system (*Di et al., 2013*; *Kim and Alger, 2010*; *Tabatadze et al., 2015*), whereas 2-AG mediates essentially all forms of phasic eCB signaling in the CNS (*Katona and Freund, 2012*; *Morena et al., 2016*). More so, as AEA mediates tonic inhibition of GABA release within the hippocampus (*Di et al., 2013*; *Lee et al., 2015*; *Tabatadze et al., 2015*), these data indicate that PAK1 disruption results in an increase in AEA production and a consequential reduction in tonic GABA release.

Quite interestingly, a recent report has indicated that AEA and 2-AG signaling within the hippocampus may compete with each other, such that elevations in AEA signaling dampen 2-AG regulation of GABAergic transmission, through an AEA-TRPV1 mediated mechanism (*Lee et al., 2015*). To investigate whether this interaction occurs in PAK1 KO mice, we analyzed phasic suppression of

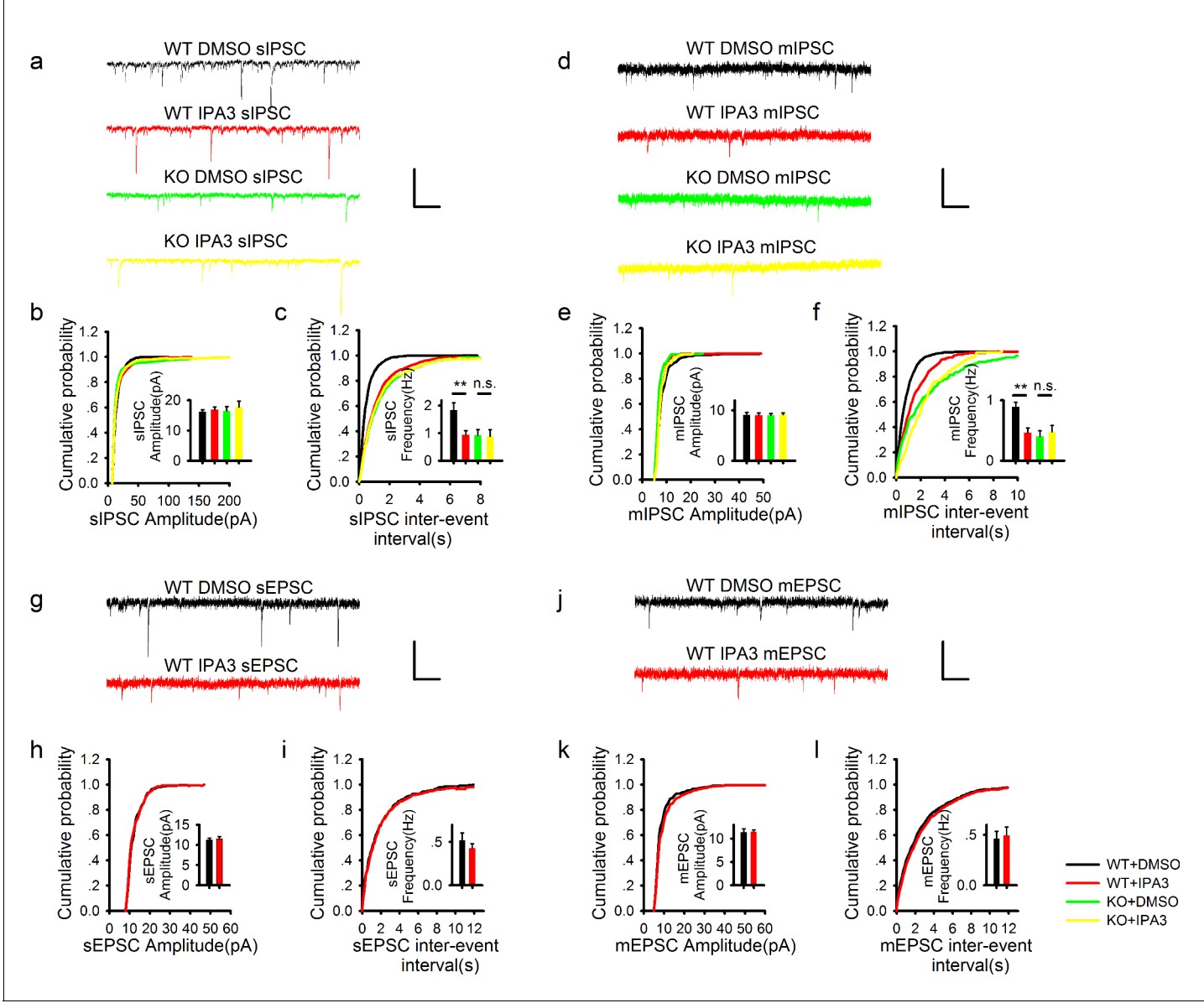

**Figure 3.** Acute disruption of postsynaptic PAK1 also selectively impairs the frequency, but not the amplitude of inhibitory synaptic responses. (a) Sample traces of sIPSC recordings. Scale bar: 60 pA/1 s. (b, c) Summary graphs of (a) showing normal amplitude (b: WT+DMSO = 16.14 ± 0.65 pA, n = 13 (4); WT+IPA3 = 16.83 ± 0.95 pA, n = 15 (4); KO+DMSO = 16.46 ± 1.41 pA, n = 9 (3); KO+IPA3 = 17.74 ± 1.98 pA, n = 9 (3); genotype: $F_{(1, 42)}$ = 0.250, p=0.620; drug: $F_{(1, 42)}$ = 0.646, p=0.426; two-way ANOVA), but reduced frequency (c: WT+DMSO = 1.84 ± 0.26 Hz, n = 13 (4); WT+IPA3 = 0.95 ± 0.15 Hz, n = 15 (4); KO+DMSO = 0.93 ± 0.20 Hz, n = 9 (3); KO+IPA3 = 0.88 ± 0.24 Hz, n = 9 (3); genotype: $F_{(1, 42)}$ = 4.908, *p=0.032; drug: $F_{(1, 42)}$ = 4.524, *p=0.039; also see *Figure 3—source data 1* for t-tests between groups) of sIPSCs in the PAK1 inhibitor IPA3 treated compared to vehicle (DMSO) treated WT neurons. (d) Sample traces of mIPSC recordings. (e, f) Summary graphs of (d) showing normal amplitude (e: WT+DMSO =9.10 ± 0.47 pA, n = 15 (6); WT+IPA3 = 9.02 ± 0.46 pA, n = 11 (6); KO+DMSO = 9.02 ± 0.36 pA, n = 10 (4); KO+IPA3 = 9.13 ± 0.36 pA, n =8 (3); genotype: $F_{(1, 40)}$ = 0.001, p=0.969; drug: $F_{(1, 40)}$ = 0.001, p=0.978; two-way ANOVA), but decreased frequency of mIPSCs (f: WT+DMSO = 0.89 ± 0.07 Hz, n = 15 (6); WT+IPA3 = 0.46 ± 0.08 Hz, n = 11 (6); KO+DMSO = 0.40 ± 0.09 Hz, n = 10 (4); KO+IPA3 = 0.48 ± 0.10 Hz, n = 8 (3); genotype: $F_{(1, 40)}$ = 7.703, **p=0.008; drug: $F_{(1, 40)}$ = 4.259, *p=0.046, two-way ANOVA; also see *Figure 3—source data 2* for t-tests between groups) in IPA3 treated compared DMSO treated WT neurons. (g) Sample traces of sEPSC recordings. (h, i) Summary graphs of (g) showing that IPA3 had no effect on either amplitude (h: WT+DMSO = 11.21 ± 0.49 pA, n = 8 (4); WT+IPA3 = 11.52 ± 0.55 pA, n = 12 (5); p=0.693; t-test) or frequency (i: WT+DMSO = 0.52 ± 0.08 Hz, n = 8 (4); WT+IPA3 = 0.43 ± 0.05 pA, n = 12 (5); p=0.322; t-test) compared to DMSO treated WT neurons. (j) Sample traces of mEPSC recordings. (k, l) Summary graphs of (j) showing that IPA3 had no effect on either amplitude (k: WT+DMSO = 11.40 ± 0.71 pA, n = 9 (5); WT+IPA3 = 11.52 ± 0.39 pA, n = 10 (5); p=0.882; t-test) or frequency (l: WT+DMSO =0.46 ± 0.07 Hz, n = 9 (5); WT+IPA3 = 0.50 ± 0.08 pA, n = 10 (5); p=0.754; t-test) of mEPSCs compared to DMSO treated WT neurons. Scale bars for **d, j** and **l**: 20 pA/1 s.

*Figure 3 continued on next page*

*Figure 3 continued*

The following source data is available for figure 3:

**Source data 1.** Statistical data summary for *Figure 3b,c*: Effect of IPA3 on frequency and amplitude of sIPSC of WT and PAK1 KO neurons using two-way ANOVA.

**Source data 2.** Statistical data summary for *Figure 3e,f*: Effect of IPA3 on frequency and amplitude of mIPSC of WT and PAK1 KO neurons using two-way ANOVA.

inhibition induced by postsynaptic depolarization (DSI) and found that the amplitude of DSI was significantly reduced in the KO compared to WT mice (*Figure 7c*), supporting the idea that elevated AEA competes with 2-AG, resulting in reduced phasic 2-AG signaling. Together, these data create a compelling argument that disruption of PAK1 selectively augments tonic AEA signaling to dampen constitutive synaptic GABA transmission.

To determine the mechanism by which PAK1 modulates the levels of eCBs, we examined how its deletion impacted a series of enzymes known to be involved in eCB metabolism. Consistent with the fact that there were no changes in 2-AG, the enzymes involved in the generation and metabolism of 2-AG, including monoacylglycerol lipase (MGL) and diacylglycerol lipase (DGL), were not altered in PAK1 KO mice (*Figure 7d–g*). Surprisingly, however, both total and synaptosomal protein levels of the enzyme primarily involved in AEA metabolism, fatty acid amide hydrolase (FAAH), were also not altered in PAK1 KO mice (*Figure 7h,i*). As several reports have identified that COX-2 can be an important regulator of AEA signaling (*Glaser and Kaczocha, 2010*; *Hermanson et al., 2013*), we examined if PAK1 deletion could impact AEA signaling through a COX-2 mediated mechanism. Notably, we found that although the total protein level of COX-2 was not altered (*Figure 7j*), its synaptosomal fraction was significantly reduced in the KO brain (*Figure 7k*, *Figure 7—figure supplement 1*), suggesting that PAK1 is specifically important for COX-2 expression at the synapse and/or during synaptic activity. Together these results suggest that the enhanced eCB signaling, due to the disruption of PAK1, might be mediated by reduced COX-2 and subsequently elevated AEA.

## PAK1 restricts tonic AEA signaling through COX-2

To directly test if the reduced COX-2 expression is responsible for the elevated effect of the eCB signaling on GABA transmission in PAK1 KO mice, we tested the effect of the COX-2 inhibitor Nimesulide on eIPSC. As shown in *Figure 8a*, application of Nimesulide (30 µM) in WT neurons depressed IPSCs to approximately 45% of the baseline response, presumably due to increased eCB production, but this Nimesulide-induced depression was significantly reduced (to 70% of the baseline response) in PAK1 KO neurons, consistent with an already reduced COX-2 level in the KO mice. Following Nimesulide application and after the depressed responses became stabilized, we then treated the neurons with the CB1 receptor antagonist AM251 (5 µM). As shown in *Figure 8b*, both WT and PAK1 KO neurons now showed equally enhanced responses (250% of the baseline response) that were similar to PAK1 KO neurons treated with AM251 alone as shown in *Figure 6a*. Therefore, inhibition of COX-2 in WT neurons recapitulated the effect of PAK1 disruption.

As COX-2 only metabolizes a proportion of AEA, changes in COX-2 are consistent with the magnitude of AEA changes we documented in the hippocampus, which are significantly less dramatic than what would be seen following disruption of FAAH activity (*Cravatt et al., 2001*). Finally, to provide further evidence that this increase in tonic eCB signaling in PAK1 KO mice is mediated by a selective increase in AEA, and not 2-AG, signaling, we examined the effects of specific AEA and 2-AG hydrolysis inhibitors. Similar to what was seen following COX-2 inhibition, the ability of the FAAH inhibitor URB597 (1 µM) to reduce eIPSC was reduced in PAK1 KO mice relative to WT mice (*Figure 8c*) and the effect of AM251 (5 µM) on eIPSC was no longer different between the two genotypes after the FAAH treatment (*Figure 8d*), consistent with the fact that there is already an elevated AEA tone. Importantly, the effect of the MGL inhibitor JZL184 was not altered in PAK1 KO mice; that is, bath application of JZL184 (5 µM) depressed eIPSC to the same degree in both WT and KO neurons (*Figure 8e*) and had no effects on the degree of disinhibition by AM251 (5 µM) (*Figure 8f*). Therefore, we conclude that the increased inhibition of GABA transmission by eCB

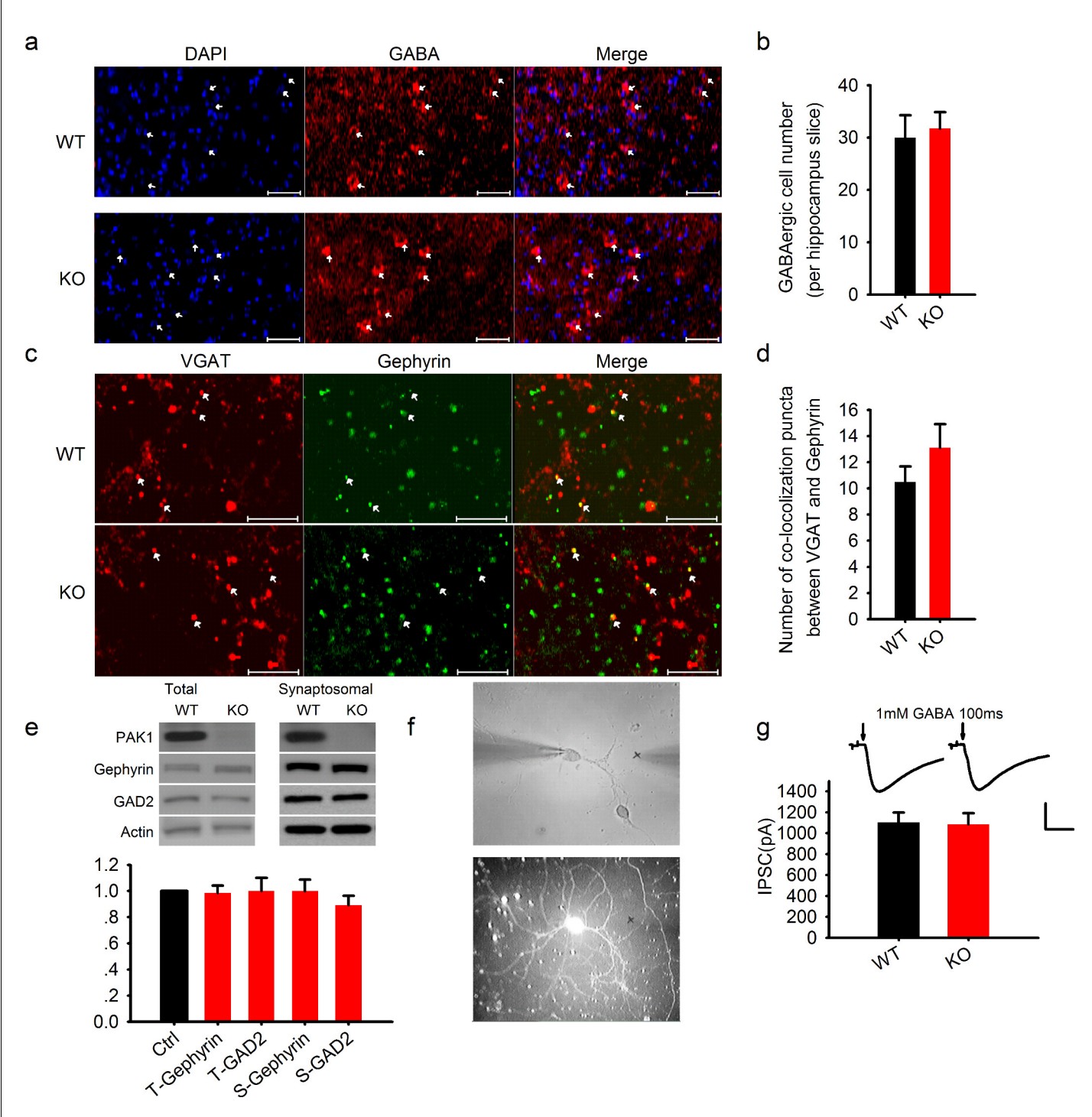

**Figure 4.** Normal GABAergic neurons, synapses, GABA receptor function and postsynaptic actin network in PAK1 KO mice. (**a**) Confocal images of hippocampal sections stained with the nucleus marker DAPI and GABA and summary graph (**b**) showing similar number of GABAergic neurons in PAK1 KO and WT control mice (WT = 30 ± 4.3 neurons/section, n = 8 (3); KO = 31.73 ± 3.1 6 neurons/section, n = 11 (4); p = 0.744; t-test). Scale bar: 100 μm. (**c**) Confocal images of hippocampal sections stained with the GABAergic presynaptic marker VGAT and postsynaptic marker gephyrin and summary graph (**d**) showing normal synapse number in PAK1 KO mice (WT = 10.50 ± 1.18 puncta/image, n = 8 (4); KO = 13.13 ± 1.79, n = 8 (4); p=0.241; t-test). Scale bar: 10 μm. (**e**) Western blots of hippocampal lysate and summary graph showing no differences in the level of total (T) and synaptosomal (S) GAD2 and gephyrin between PAK1 KO and WT control mice (KO T-gephyrin = 0.99 ± 0.06, n = 7 (7), p=0.797; T-GAD2 = 1.00 ± 0.10, n = 8 (8), p=0.990; S- gephyrin = 1.00 ± 0.09, n = 7 (7), p=0.986; S-GAD2 = 0.89 ± 0.07, n = 7 (7), p=0.158; t-tests). (**f**) Phase contrast (upper) and GFP (lower) images of

*Figure 4 continued on next page*

*Figure 4 continued*

cultured hippocampal neurons showing how the genotype of mixed neurons was identified based on the presence or absence of GFP; (**g**) Sample traces (upper) and summary graph (lower) showing no differences in the amplitude of responses evoked by 1 mM GABA puff (arrows) delivered to the cell body of the neurons (WT = 1103.74 ± 93.43 pA, n = 18 (3); KO = 1086.86 ± 104.94 pA, n = 18 (3); p=0.91; t-test). Scale bar: 500 pA/0.5 s.

The following figure supplement is available for figure 4:

**Figure supplement 1.** Normal GABAergic neurons and synapses in the cortex.

signaling in PAK1 KO mice is caused by reduced COX-2 expression and consequential elevation in AEA, but not 2-AG, signaling at GABA synapses in the CA1 region of the hippocampus.

## Reduced COX-2 at GABAergic, but not excitatory synapses

To further elucidate why disruption of PAK1 and reduced COX-2 affected inhibitory, but not excitatory synaptic transmission, we performed immunostaining experiments using cultured hippocampal neurons to determine the subcellular distribution of PAK1 and COX-2, and how their synaptic localization was affected by PAK1 disruption. First, we showed that in WT neurons, PAK1 was colocalized with both PSD-95 (excitatory synaptic marker, *Figure 9a*) and gephyrin (GABAergic synaptic marker, *Figure 9b*), suggesting that PAK1 is expressed at both excitatory and inhibitory synapses. In addition, we showed that PAK1 was colocalized with COX-2 (*Figure 9c*). Next, we showed that a small portion of COX-2 was colocalized with PSD-95 (*Figure 9d*) and this colocalization was not altered in PAK1 KO neurons (*Figure 9e–h*), suggesting that PAK1 disruption does not affect COX-2 localization at the excitatory synapse. Finally, we showed that a much larger portion of COX-2 was colocalized with gephyrin (*Figure 9i*), and importantly, the level of this colocalization was significantly reduced in PAK1 KO compared to WT neurons (*Figure 9i–m*). The total protein levels of COX-2 (*Figure 9f,k*), PSD-95 (*Figure 9g*) and gephyrin (*Figure 9l*) were unaltered in PAK1 KO neurons. These results together suggest that PAK1 disruption specifically impairs COX-2 localization at GBABergic synapses, providing a mechanism for PAK1-COX2 signaling to specifically regulate inhibitory synaptic transmission.

## Discussion

In this study we have revealed a novel signaling pathway by which PAK1 regulates the synaptic expression of COX-2 to restrict tonic eCB signaling. Specifically, genetic or pharmacological disruption of PAK1 signaling reduces synaptic COX-2 levels and increases tonic AEA signaling to reduce constitutive GABAergic synaptic transmission in the hippocampus. Importantly, this effect of PAK1 disruption on GABA transmission is replicated by the direct application of a COX-2 inhibitor, which is also dependent on the activation of presynaptic CB1 receptors. While several reports have identified COX-2 as a regulator of eCB signaling, these data represent the first demonstration of a physiological substrate (PAK1 in this case), which can mobilize COX-2 to the synaptic compartment to modulate eCB signaling in real time. In addition, these data provide some of the first data regarding intracellular signaling mechanisms that can regulate tonic AEA signaling and, importantly, could potentially have significant implications for a wide range of pathologies that involve dysregulation of this cascade.

## PAK1 is specifically required for GABAergic synaptic transmission

Although PAK1 is known to be important in the regulation of spine properties such as spine actin and morphology, its effect on excitatory synaptic transmission is minimal (*Asrar et al., 2009*). In fact, basal excitatory synaptic transmission is not altered in PAK1 KO mice, possibly due to the existence of other functionally redundant PAKs (e.g. PAK2 and PAK3) (*Bokoch, 2003*; *Huang et al., 2011*; *Kelly and Chernoff, 2012*; *Meng et al., 2005*). In contrast, disruptions of PAK1, either by genetic deletion or pharmacological blockade, dramatically reduces GABAergic transmission, indicating that the primary role of PAK1 is to facilitate inhibitory but not excitatory synaptic function. This selective effect of PAK1 disruption is rather surprising because it has been shown previously that expression of a dominant negative form (i.e. the autoinhibitory domain) of PAK1 affects the excitatory synaptic

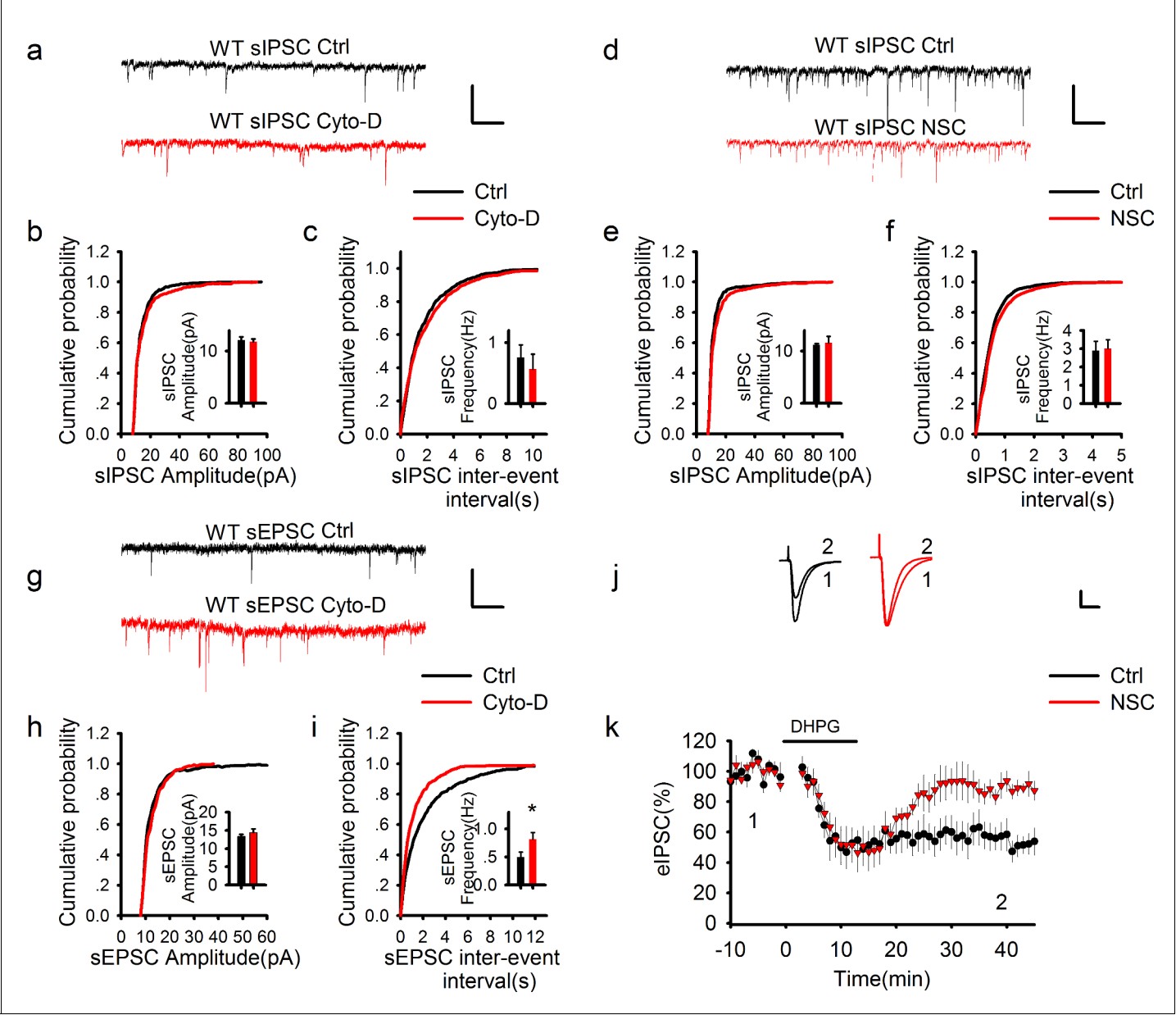

**Figure 5.** GABAergic transmission is independent of postsynaptic actin cytoskeleton. (a–f) Sample traces of sIPSCs and summary graphs showing neither the actin polymerization inhibitor cytochalasin D (a–c) nor Rac1 inhibitor NSC23766 (d–f) had an effect on the amplitude (b: DMSO = 12.09 ± 0.65pA, n = 6 (3); Cyto-D = 11.82 ± 0.54 pA, n = 6 (3); p=0.757; e: Ctrl = 11.21 ± 0.26 pA, n = 6 (3); NSC = 11.57 ± 1.29 pA, n = 7 (3); p=0.806; t-tests) or frequency (c: DMSO = 0.79 ± 0.25 Hz, n = 6 (3); Cyto-D = 0.57 ± 0.24 Hz, n = 6 (3); p=0.538; f: Ctrl = 2.90 ± 0.51 Hz, n = 6 (3); NSC = 3.01 ± 0.48 Hz, n = 7 (3); p=0.867; t-tests). Scale bar: 20 pA/1 s. (g–i) Sample traces of sEPSCs and summary graphs showing actin polymerization inhibitor cytochalasin D had no effect on the amplitude (h: Ctrl = 13.48 ± 0.91 pA, n = 8 (3); NSC = 15.48 ± 1.33 pA, n=6 (3); p=0.248; t-test), but significantly increased the frequency of eEPSCs (i: Ctrl = 0.47 ± 0.09 Hz, n = 8 (3); NSC = 0.95 ± 0.17 Hz, n = 6 (3); *p=0.034; t-test). (j, k) Representative traces and summary graph of evoked EPSC (eEPSC) showing that the Rac1 inhibitor NSC23766 blocked DHPG-induced LTD (Ctrl = 52.04 ± 7.33%, n = 5 (5); NSC = 90.03 ± 5.21%, n=5 (5); *p=0.027; t-test). Scale bar: 25 pA/25 ms.

morphology (*Hayashi et al., 2004*). One possible explanation is that the expression of this mutant PAK1 may affect other members of the PAK family, including PAK3, which is also highly expressed in the brain (*Meng et al., 2005*). Indeed, the double KO mice lacking both PAK1 and PAK3 display severe deficits in spines and the actin cytoskeleton (*Huang et al., 2011*). These results together suggest that while PAK1 and PAK3 are functionally redundant at the excitatory synapses, PAK1 is a

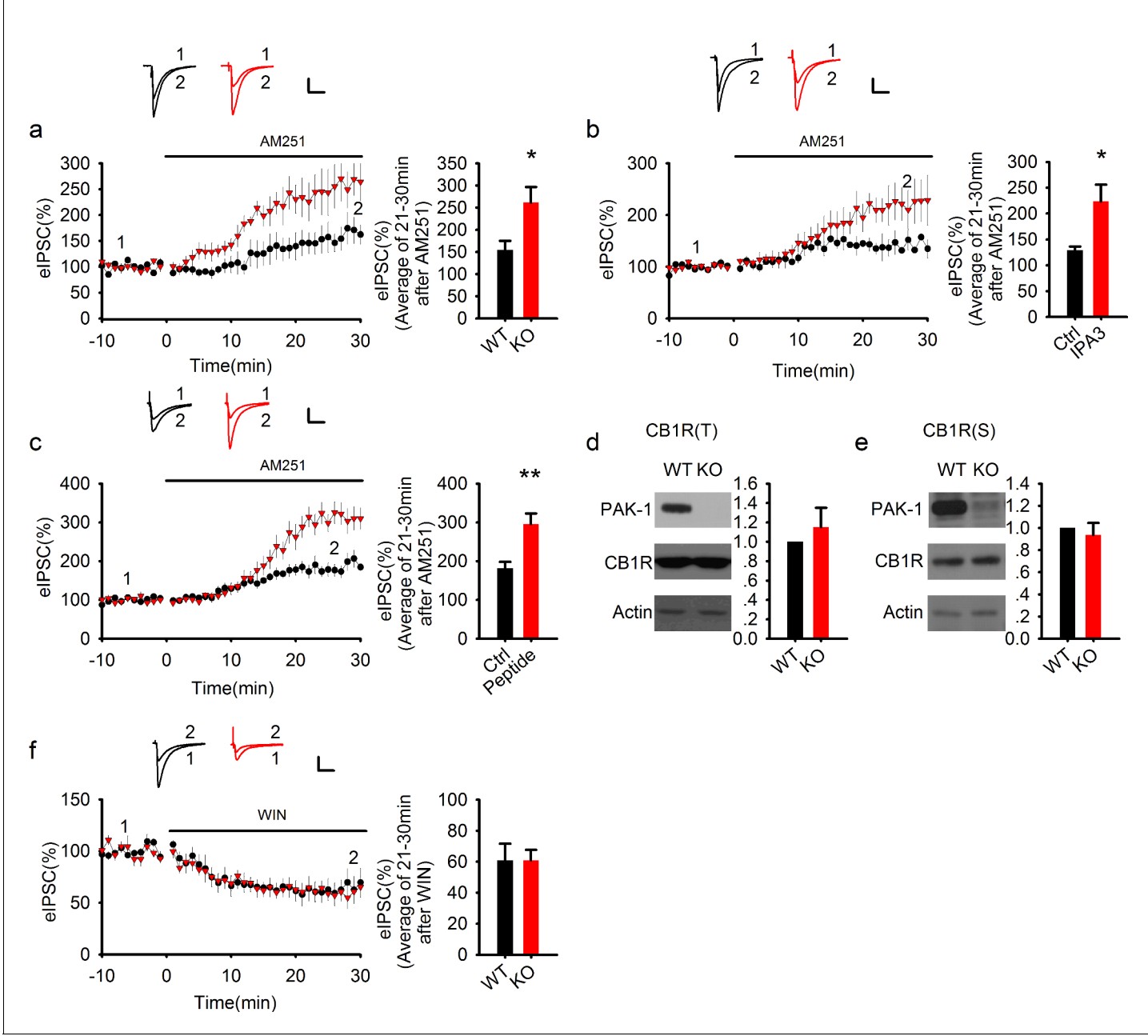

**Figure 6.** PAK1 disruption enhances endocannabinoid signaling. (a) Sample traces of eIPSC and averaged data showing that bath application of the CB1 receptor antagonist AM251 potentiated the amplitude of eIPSC significantly more in PAK1 KO compared to WT control (genotype: $F_{(1, 11)}$ = 7.51, *p=0.02; time: $F_{(3, 43)}$ = 26.833, ***p <0.001; repeated measures two-way ANOVA [also see *Figure 6—source data 1*]; at 21-30 min post AM251 application: WT = 154.77 ± 20.57%, n = 7 (5); KO = 261.41 ± 35.00%, n = 6 (4), *p=0.02; t-test). (b) Sample traces of eIPSC and averaged data showing that postsynaptic infusion of the PAK1 inhibitor IPA3 (for 30 min) was sufficient to enhance the subsequent potentiating effect of AM251 on eIPSC (genotype: $F_{(1, 11)}$ = 4.919, *p=0.049; time: $F_{(3, 33)}$ = 15, p<0.001; repeated measures two-way ANOVA [also see *Figure 6—source data 2*]; at 21-30 min post AM251 application: Ctrl = 129.17 ± 7.20%, n = 6 (5); IPA3 = 227.26 ± 38.26%, n = 7 (5); *p=0.024; t-test). Baseline responses (-10-0 min) shown here were taken 30 min after whole-cell break-in. (c) Sample traces of eIPSC and averaged data showing that postsynaptic infusion of the PAK1 inhibitory peptide (for 30 min) was sufficient to enhance the subsequent potentiating effect of AM251 on eIPSC (genotype: $F_{(1, 11)}$ = 10.254, **p=0.008; time: $F_{(3, 33)}$ = 70.824, ***p<0.001; repeated measures two-way ANOVA [also see *Figure 6—source data 3*]; at 21–30 min post AM251 application: Ctrl = 181.89 ± 15.94%, n = 7 (5); peptide = 295.71 ± 27.40%, n = 6 (4); ***p=0.001; t-test). Baseline responses (-10-0 min) shown here were taken 30 min after whole-cell break-in. (d, e) Western blots of hippocampal lysate and summary graphs showing that both total (d) and synaptosomal (e) CB1R protein levels were unaltered in PAK1 KO mice (KO total = 1.15 ± 0.20, n = 7 (7); p=0.468 normalized and compared to the WT; KO synaptosomal = 0.94 ± 0.11, n = 8 (8); p=0.567 normalized and compared to the WT; t-tests). (f) Sample traces of eIPSC and averaged data showing that bath application of the CB1 receptor agonist WIN depressed eIPSC to the same degree in PAK1 KO and WT control (genotype: $F_{(1, 8)}$ = 0.008, p=0.932;

*Figure 6 continued on next page*

*Figure 6 continued*

time: F(3, 24) = 31.556, ***p<0.001; repeated measures two-way ANOVA [also see *Figure 6—source data 4*]; at 21-30 min post WIN application: WT = 60.78 ± 10.78%, n = 5 (5); KO = 60.88 ± 6.73%, n = 5 (5); p=0.977; t-test). All scale bars: 40 pA/25 ms.

The following source data and figure supplements are available for figure 6:

**Source data 1.** Statistical data summary for *Figure 6a*: Effect of AM251 on eIPSC in WT and PAK1 KO using repeated measures two-way ANOVA.

**Source data 2.** Statistical data summary for *Figure 6b*: Effect of AM251 on eIPSC in the presence or absence of IPA3 using repeated measures two-way ANOVA.

**Source data 3.** Statistical data summary for *Figure 6c*: Effect of AM251 on eIPSC in the presence or absence of PAK1 inhibitory peptide using repeated measures two-way ANOVA.

**Source data 4.** Statistical data summary for *Figure 6f*: Effect of WIN on eIPSC in WT and PAK1 KO using repeated measures two-way ANOVA.

**Figure supplement 1.** The lack of effect of DMSO on eIPSCs.

**Figure supplement 1—source data 1.** Statistical data summary for *Figure 6—figure supplement 1*: Lack of effect of DMSO on eIPSC using repeated measures two-way ANOVA.

unique regulator at the inhibitory synapses, which may not be readily replaced by other members of the PAK family. Therefore, the ability of the PAK1 inhibitors in ameliorating the deficits associated with models of fragile X syndrome and neurofibromatosis (*Dolan et al., 2013*; *Hayashi et al., 2007*; *Molosh et al., 2014*) may be mediated by its effect on the GABA function. Consistent with this notion, inhibitory synaptic transmission is commonly altered in these brain disorders (*Cui et al., 2008*; *Olmos-Serrano et al., 2010*; *Radhu et al., 2015*).

## Postsynaptic PAK1 regulates presynaptic GABA release through the eCB system

In PAK1 KO mice, the frequency, but not the amplitude of mIPSCs and sIPSCs is reduced. The reduction in the frequency is not caused by changes in the number of inhibitory neurons or synapses because none of these parameters are altered in the KO mice. The lack of changes in synapse number is also consistent with the observations that the frequency reduction can be rapidly induced (within mins) by infusion of the PAK1 inhibitors, a perturbation that is not likely to cause any changes in the number of neurons or synapses. Our data also indicate that although the frequency reduction is induced by postsynaptic inhibition of PAK1, it is independent of postsynaptic GABA receptors or the actin cytoskeleton. Therefore, postsynaptic PAK1 affects GABA transmission through a retrograde messenger to modulate neurotransmitter release. Several lines of evidence support that eCBs are the retrograde messenger to mediate such an effect here. First, the effect the CB1 receptor antagonist AM251 is greatly enhanced by the loss of PAK1; application of AM251 potentiates eIPSC three times greater in PAK1 KO or IPA3 treated neurons than in WT neurons. The simplest interpretation of this result is that loss of PAK1 results in increased tonic eCB signaling and therefore a greater inhibition of GABA release. Thus, blocking eCB inhibition causes a greater potentiation of GABA release and eIPSC in the absence of PAK1. There are at least two possibilities by which the eCB signaling can be enhanced: an increase in the signaling pool of eCB molecules and/or altered signaling processes triggered by activation of the CB1 receptors. Because neither the expression level of CB1 receptors nor the effect of a CB1 receptor agonist are enhanced, a straightforward interpretation would be that eCB secretion is elevated in the absence of PAK1. Consistent with this, biochemical analysis of hippocampi from PAK1 KO mice indicate a significant increase in the tissue level of the eCB AEA, but not 2-AG. While 2-AG has been found to mediate the majority of phasic eCB processes, such as DSI and LTD, AEA is believed to mediate the tonic signaling actions of the eCB system (*Katona and Freund, 2012*; *Morena et al., 2016*). As such, the selective increase in AEA is consistent with the fact that tonic, but not phasic, eCB signaling is enhanced by disruption of PAK1. In fact, a recent report has suggested that elevations in AEA signaling reduce the ability of 2-AG signaling to modulate GABAergic transmission in the hippocampus (*Lee et al., 2015*), and again

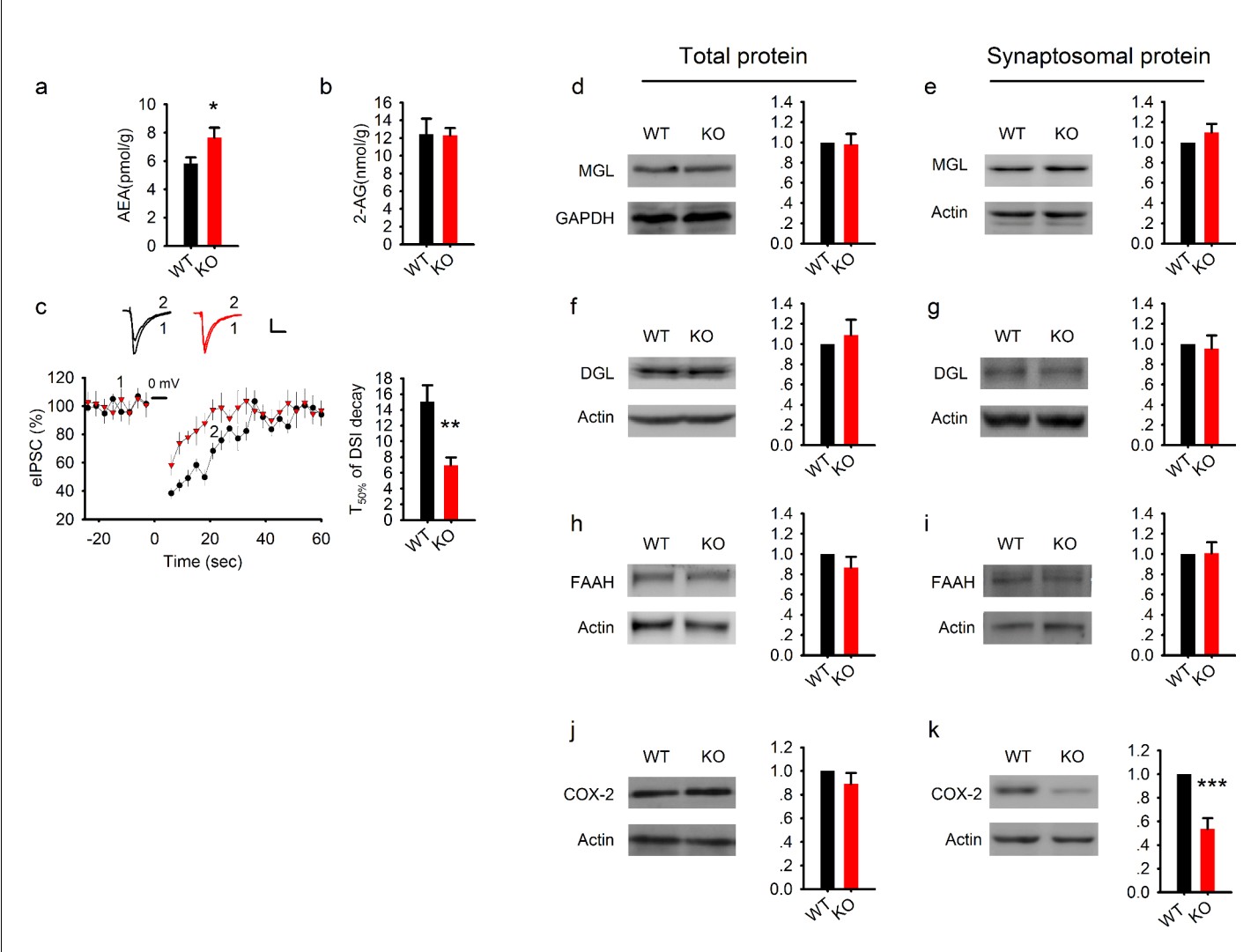

**Figure 7.** Elevated AEA and reduced COX-2 in PAK1 KO mice. (a) Summary graph showing a significant increase in hippocampal tissue AEA in PAK1 KO compared to WT control (WT = 5.82 ± 0.44 pmol/g, n = 11 (11); KO = 7.64 ± 0.70 pmol/g, n = 13 (13), *p=0.046; t-test). (b). Summary graph showing no differences in hippocampal tissue 2-AG between PAK1 KO and WT control (WT = 12.46 ± 1.73 nmol/g, n = 11 (11); KO = 12.30 ± 0.82 nmol/g, n = 13 (13), p=0.932; t-test). (c) Sample traces of eIPSC and averaged data showing reduced DSI in PAK1 KO compared to WT control (time to reach 50% of baseline response T50%: WT = 15.06 ± 2.06s, n = 9 (5); KO = 6.98 ± 0.99s, n = 9 (3); **p=0.003; t-test). (d) Western blots of hippocampal lysate and summary graphs showing no differences in total MGL protein levels between PAK1 KO and WT control (KO = 0.98 ± 0.10, n = 6 (6), p=0.854 normalized and compared to WT; t-test). (e) Western blots of hippocampal synaptosomal protein fraction and summary graph showing no difference in the amount of MGL protein between PAK1 KO and WT control (KO = 1.01 ± 0.09, n = 7 (7); p=0.253 normalized and compared to WT; t-test). (f) Western blots of hippocampal lysate and summary graph showing no differences in total DGL protein levels between PAK1 KO and WT control (KO = 1.09 ± 0.15, n = 6 (6), p=0.583 normalized and compared to WT; t-test). (g) Western blots of hippocampal synaptosomal protein fraction and summary graph showing no difference in the amount of DGL protein between PAK1 KO and WT control (KO = 0.96 ± 0.13, n = 9 (9); p=0.738 normalized and compared to WT; t-test). (h) Western blots of hippocampal lysate and summary graph showing no differences in total FAAH protein levels between PAK1 KO and WT control (KO = 0.87 ± 0.11, n = 5 (5), p=0.247 normalized and compared to WT; t-test). (i) Western blots of hippocampal synaptosomal protein fraction and summary graph showing no difference in the amount of FAAH protein between PAK1 KO and WT control (KO = 1.01 ± 0.11, n = 6 (6); p=0.937 normalized and compared to WT; t-test). (j) Western blots of hippocampal lysate and summary graphs showing no differences in total COX-2 protein levels between PAK1 KO and WT control (KO = 0.89 ± 0.09, n = 7 (7), p=0.263 normalized and compared to WT; t-test). (k) Western blots of hippocampal synaptosomal protein fraction and summary graph showing reduced COX-2 protein in PAK1 KO compared to WT control (KO = 0.54 ± 0.09, n = 10 (10); ***p<0.001 normalized and compared to WT; t-test).

The following figure supplement is available for figure 7:

**Figure supplement 1.** Reduced synaptosomal COX-2 in PAK1 KO hippocampus.

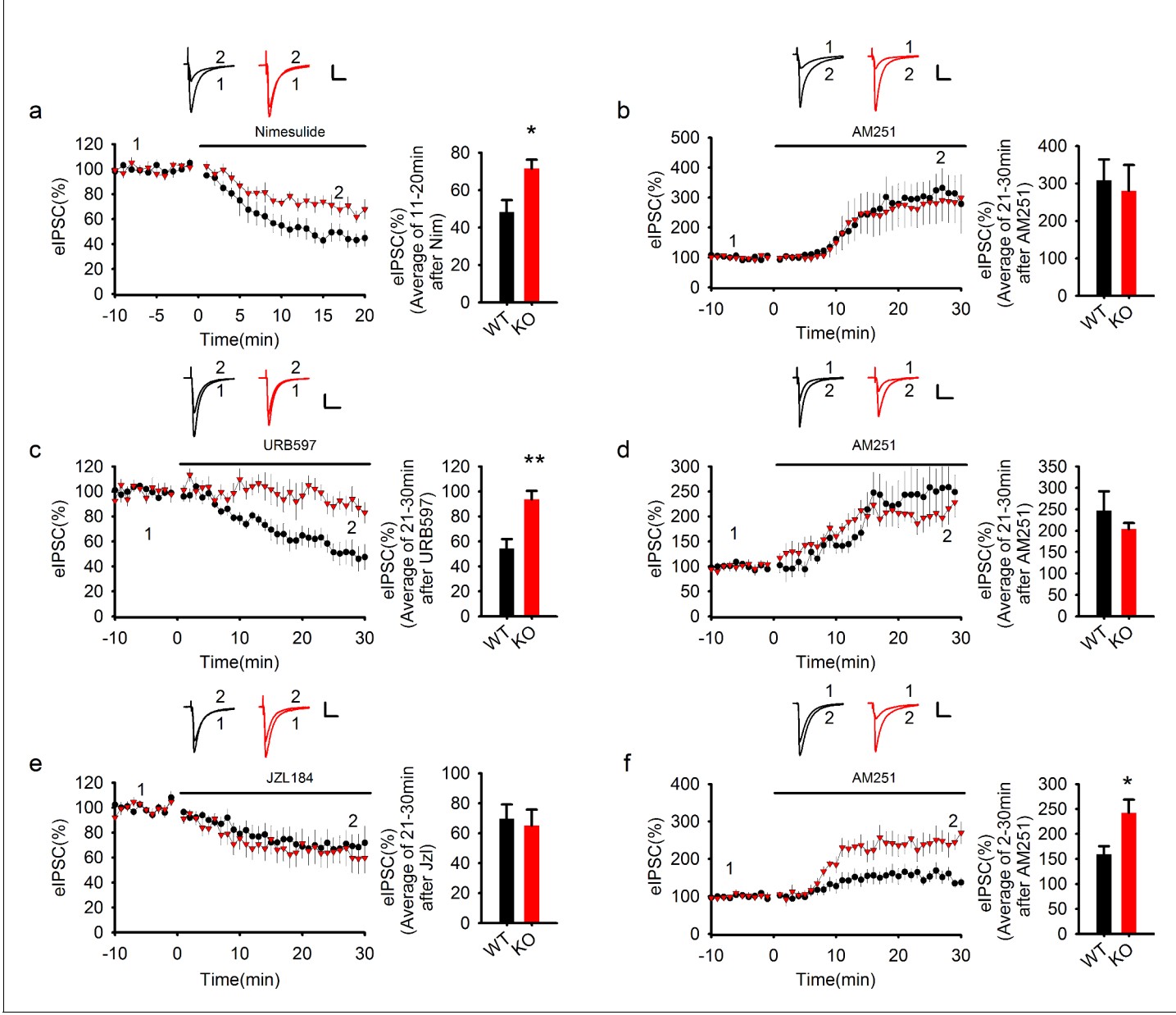

**Figure 8.** COX-2 inhibition recapitulates the effect of PAK1 disruption. (**a**) Sample traces and averaged data of eIPSCs showing that bath application of the COX-2 inhibitor Nim depressed eIPSCs in WT, but this depression was significantly reduced in PAK1 KO neurons (genotype: $F_{(1, 12)}$ = 7.639, *p=0.017; time: $F_{(2, 24)}$ = 87.676, ***p<0.001; repeated measures two-way ANOVA [also see *Figure 8—source data 1*]; at 11-20 min post Nim application: WT = 48.26 ± 6.42%, n = 7 (3); KO = 71.51 ± 4.75%, n = 7 (3); *p=0.013; t-test). Scale bar: 30 pA/25 ms. (**b**) Sample traces and averaged data showing that following the Nim treatment, AM251 potentiated eIPSCs to the same degree in WT and PAK1 KO neurons (genotype: $F_{(1, 9)}$ = 0.044, p=0.839; time: $F_{(3, 27)}$ = 15.222; ***p<0.001; repeated measures two-way ANOVA [also see *Figure 8—source data 2*]; at 21-30 min post AM251 application: WT = 308.58 ± 55.37%, n = 5 (4); KO = 280.53 ± 69.32%, n = 6 (5); p=0.766; t-test). Scale bar: 60 pA/25 ms. Baseline responses (-10-0 min) shown here were taken 30 min after the onset of the Nim treatment. Nim was present throughout the entire experiment. (**c**) Sample traces and averaged data of eIPSCs showing that bath application of the FAAH inhibitor URB597 depressed eIPSCs in WT, but this depression was significantly reduced in PAK1 KO neurons (genotype: $F_{(1, 13)}$ = 13.830, **p=0.003; time: $F_{(3, 39)}$ = 14.122, ***p<0.001; repeated measures two-way ANOVA [also see *Figure 8—source data 3*]; at 21-30 min post URB597 application: WT = 54.19 ± 7.59%, n = 7 (5); KO = 93.74 ± 6.80%, n = 8 (3); **p=0.002; t-test). Scale bar: 70 pA/25 ms. (**d**) Sample traces and averaged data showing that following the URB597 treatment, AM251 potentiated eIPSCs to the same degree in WT and PAK1 KO neurons (genotype: $F_{(1, 8)}$ = 0.055, p=0.821; time: $F_{(3, 24)}$ = 23.459, ***p<0.001; repeated measures two-way ANOVA [also see *Figure 8—source data 4*]; at 21-30 min post AM251 application: WT = 246.61 ± 45.34%, n = 5 (3); KO = 208.59 ± 17.52%, n = 5 (4); p=0.391; t-test). Scale bar: 60 pA/25 ms. Baseline responses (-10-0 min) shown here were taken 30 min after the onset of the URB597 treatment. URB597 was present throughout the entire experiment. (**e**) Sample traces and averaged data showing no differences in eIPSC depression by the MGL inhibitor JZL184 between PAK1 KO and WT control (genotype: $F_{(1, 16)}$ = 0.265, p=0.614; time: $F_{(3, 48)}$ = 17.292, ***p<0.001; repeated measures two way ANOVA [also

*Figure 8 continued on next page*

*Figure 8 continued*

see *Figure 8—source data 5*]; at 21-30 min post JZL184 application: WT = 69.53 ± 9.60%, n = 10 (8); KO = 64.99 ± 10.76%, n = 8 (6); p=0.757; t-test). Scale bar: 35 pA/25 ms. (**f**) Sample traces and averaged data showing that following JZL184 treatment, AM251 still induced eIPSC potentiation significantly more in PAK1 KO compared to WT control (genotype: $F_{(1, 9)}$ = 7.770, *p=0.021; time: $F_{(3, 27)}$ = 30.146, ***p<0.001; repeated measures two-way ANOVA [also see *Figure 8—source data 6*]; at 21–30 min post AM251 application: WT = 159.76 ± 16.08%, n = 6 (6); KO = 242.60 ± 26.03%, n = 5 (4); *p=0.020; t-test). Baseline responses (-10-0 min) shown here were taken 30 min after the onset of the JZL184 treatment. JZL184 was present throughout the entire experiments. Scale bar: 60 pA/25 ms.

The following source data is available for figure 8:

**Source data 1.** Statistical data summary for *Figure 8a*: Effect of Nimesulide on eIPSC in WT and PAK1 KO using repeated measures two-way ANOVA.

**Source data 2.** Statistical data summary for *Figure 8b*: Effect of AM251 on eIPSC after Nimesulide treatment in WT and PAK1 KO using repeated measures two-way ANOVA.

**Source data 3.** Statistical data summary for *Figure 8c*: Effect of URB597 on eIPSC in WT and PAK1 KO using repeated measures two-way ANOVA.

**Source data 4.** Statistical data summary for *Figure 8d*: Effect of AM251 on eIPSC after URB597 treatment in WT and PAK1 KO using repeated measures two-way ANOVA.

**Source data 5.** Statistical data summary for *Figure 8e*: Effect of JZL184 on eIPSC in WT and PAK1 KO using repeated measures two-way ANOVA.

**Source data 6.** Statistical data summary for *Figure 8f*: Effect of AM251 on eIPSC after JZL treatment in WT and PAK1 KO using repeated measures two-way ANOVA.

consistent with this finding, our data demonstrate that DSI, a measure of activity-dependent, phasic eCB signaling that is mediated by 2-AG, is reduced in PAK1 KO mice. Finally, PAK1 KO mice exhibit no alteration to the synaptic response of a 2-AG hydrolysis inhibitor, but do demonstrate reduced responses to a AEA-hydrolysis inhibitor, which is to be expected if PAK1 deletion selectively increases AEA, and not 2-AG, signaling. Taken together, these results suggest that PAK1 normally restricts tonic AEA level to enhance GABA release and maintains sufficient inhibitory transmission.

## PAK1 regulates eCB signaling by targeting COX-2

The disruption of PAK1 could increase AEA signaling through a variety of mechanisms, although if AEA synthesis were increased by PAK1 deletion, then the effects of an AEA hydrolysis inhibitor would be expected to be amplified, not impaired, because the inhibitor would lock all of the excess AEA within the synapse. If PAK1 disruption reduced AEA hydrolysis, however, one would expect an occlusion to the addition of an AEA-hydrolysis inhibitor. As the latter, and not the former, is what the current data indicate, this leads us to investigate how PAK1 was modulating AEA clearance, not synthesis. FAAH is the primary enzyme responsible for AEA hydrolysis, however our biochemical analysis did not reveal any difference between total or synaptic expression of FAAH protein in PAK1 KO mice. A growing body of work has identified that aside from the canonical metabolism of AEA by FAAH, COX-2 represents an additional mechanism of AEA clearance (*Glaser and Kaczocha, 2010*; *Hermanson et al., 2013*). Notably, the impact of COX-2 inhibition on AEA levels (*Hermanson et al., 2013*) is roughly comparable to the increase we documented herein in PAK1 KO mice. Consistent with this, we found that PAK1 KO mice did exhibit reductions in COX-2 protein. It is important to emphasize that only synaptic, but not total COX-2 is affected in PAK1 KO mice, implying that PAK1 is particularly important for COX-2 regulation at the synapse. Furthermore, although COX-2 is expressed at both excitatory and inhibitory synapses, the effect of PAK1 deletion on COX-2 appears to be specific to inhibitory synapses, which may contribute to the specific effect of PAK1-COX-2 signaling on inhibitory synaptic transmission. The mechanism by which PAK1 regulates COX-2 localization is unknown, but it is possible that local regulation of COX-2, both at the level of protein synthesis and/or trafficking, could be targeted by PAK1.

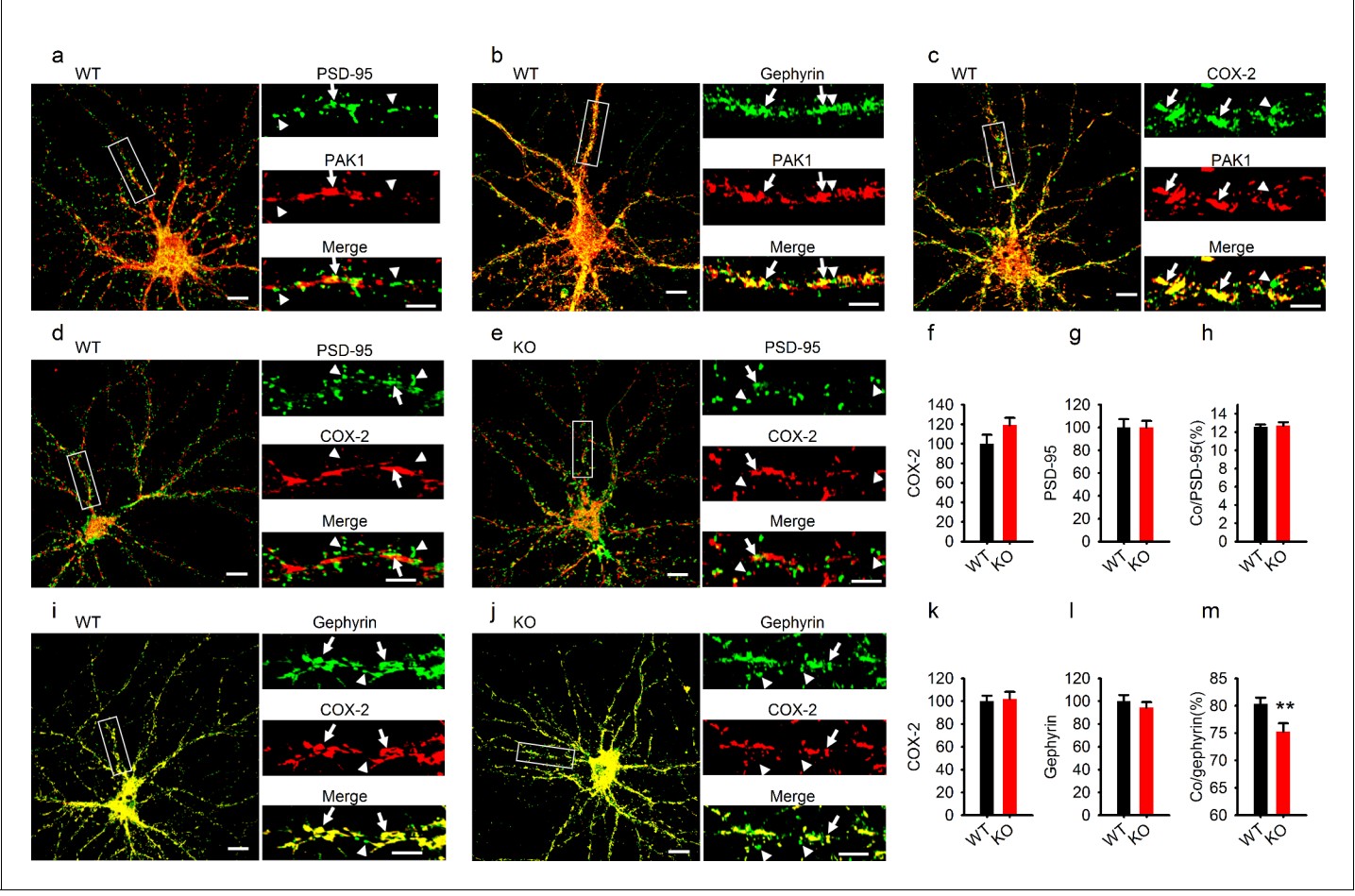

**Figure 9.** Reduced COX-2 localization at GABAergic synapses in PAK1 KO neurons. (**a–c**) Cultured hippocampal neurons costained for PAK1 and the excitatory marker PSD-95 (**a**), the GABAergic marker gephyrin (**b**) or COX-2 (**c**) showing PAK1 colocalization with PSD-95, gephyrin and COX-2. (**dh**) Cultured hippocampal neurons costained for COX-2 and PSD9-5 in WT (**d**) and PAK1 KO neurons (**e**) and summary graphs (**fh**) showing no differences between genotypes in total COX-2 (**f**, WT = 100 ± 9.11, n = 21 (3); KO = 119.08 ± 7.50, n = 16 (3); p=0.131; t-test), total PSD-95 (**g**, WT = 100 ± 7.33, n = 21 (3); KO = 100.03 ± 5.68, n = 16 (3); p=0.997; t-test) and COX-2 colocalized with PSD-95 (**h**, WT = 12.55 ± 0.26%, n = 21 (3); KO = 12.72 ± 0.35%, n = 16 (3); p=0.700; t-test). (**im**) Cultured hippocampal neurons costained for COX-2 and gephyrin in WT (**i**) and PAK1 KO neurons (**j**) and summary graphs (**km**) showing no changes in total COX-2 (**k**, WT = 100 ± 4.90, n = 17 (3); KO = 102.09 ± 6.14, n = 15 (3); p=0.792; t-test) or total gephyrin (**l**, WT = 100 ± 5.31, n = 17 (3); KO = 94.29 ± 4.91, n = 15 (3); p=0.432; t-test), but reduced COX-2 colocalized with gephyrin (**m**, WT = 80.35 ± 1.14, n = 17 (3); KO = 75.32 ± 1.47, n = 15 (3); **p=0.008; t-test). Scale bars: 10 μm for whole neuron images and 5 μm for the enlarged dendritic fragments. Arrows indicate colocalization and arrowheads for no colocalization.

## Implications and conclusions

With respect to neurodevelopmental disorders, such as autism, these data could have significant implications for both the therapeutic potential of PAK1 inhibitors or agents that enhance AEA signaling. For example, PAK1 deletion has been shown to normalize synaptic and behavioral deficits in the Neurofibromatosis model of autism (*Molosh et al., 2014*). Specifically, deletion of PAK1 resulted in a reduction of mIPSC in the amygdala (similar to what was found within the hippocampus within the current study) that was associated with an improvement in social behaviors (*Molosh et al., 2014*). In line with this, elevating AEA signaling within the amygdala (which has been shown to dampen GABA release in the amygdala [*Azad et al., 2004*]) has been shown to improve social interaction and increase social behavior (*Trezza et al., 2012*). More so, a recent report has indicated that the neuroligin-3 mutation related to autism causes a disruption in tonic eCB signaling within the hippocampus (*Földy et al., 2013*), resulting in an increase in GABA release and a shift in the E/I balance of the hippocampus that is the exact opposite of what was produced by PAK1 disruption. Finally, PAK1

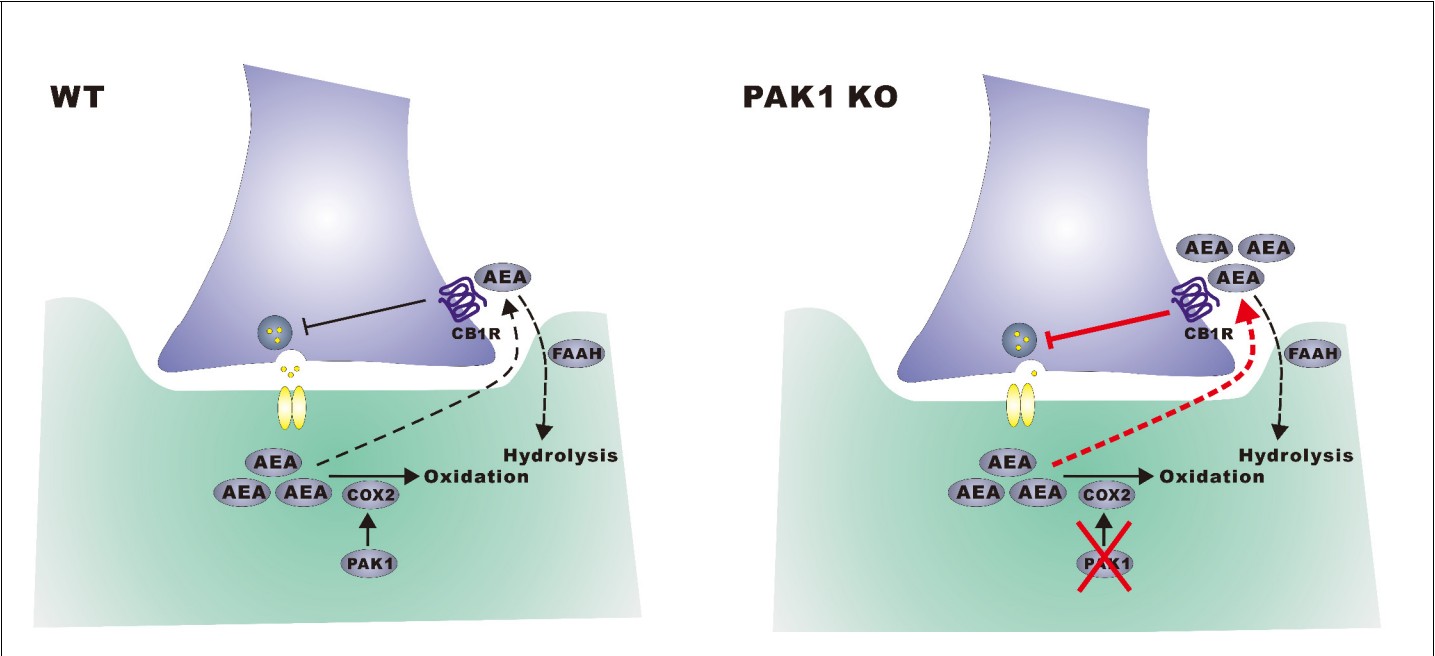

**Figure 10.** A hypothetical model. In wild type neurons, constitutively active PAK1 is required for maintaining a sufficient level of synaptic COX-2 to keep AEA low, thus less suppression of GABA release and normal inhibitory transmission. In the absence of PAK1, synaptic COX-2 is reduced, which leads to accumulation of AEA, increased suppression of GABA release and impaired inhibitory transmission and E/I balance.

disruption has also been shown to normalize behavioral changes in the Fragile X model of autism (*Dolan et al., 2013*; *Hayashi et al., 2007*), which parallels recent behavioral work similarly demonstrating that administration of AEA, but not 2-AG, hydrolysis inhibitors, can normalize some of the same behavioral deficits seen in Fragile X mice (*Qin et al., 2015*). Taken together, while speculative at this time, these data collectively create a compelling picture that impairments in tonic AEA signaling may relate to the pathology of neurodevelopmental disorders, such as autism, and that inhibition of PAK1 may exert its potentially beneficial effects by enhancing tonic AEA signaling. Future research will need to investigate this mechanism more directly, but these data establish a framework to approach this question. More so, as PAK1 has been associated with other disease states (*Gilman et al., 2011*; *Kelly and Chernoff, 2012*; *Kumar et al., 2006*; *Ma et al., 2012*), the relationship between PAK1, COX-2 and AEA signaling could prove to be highly relevant for a wide array of pathological processes, given that COX-2 and AEA have similarly been implicated in the etiology of these disease processes (*Hermanson et al., 2014*).

In summary, we have identified a novel process by which PAK1 regulates the eCB system and inhibitory synaptic function (*Figure 1*). Given that PAK1 is involved in both normal physiological and pathological processes as discussed above, that range from cancers, allergies to mental disorders, our results provide a new mechanism and treatment scheme by which PAK operates in these various systems, and opens the door to a mechanism-driven therapeutic approach which targets the interaction of these systems.

## Materials and methods

### Animals and chemicals

The generation and initial characterization of PAK1 KO mice were described previously (*Asrar et al., 2009*; *Huang et al., 2011*). All the mice used in this study were PAK1 KO (*Pak1-/-*) offspring and their wild type littermaze (*Pak1+/+*) derived PAK1 heterozygous (*Pak1+/-*) breeders to minimize the effect of genetic or environmental factors. The age of the animals in all experiments was 25–32 days. The following PCR primers were used for PAK1 mouse genotyping: (*Pak1*+F: 5'-C

TGAGGGAAGAGACTGCAGAG-3', *Pak1*+R: 5'-AGGCAGAGGTTTGGAGCCGTG-3'; *Pak1*-F: 5'-C TGAGGGAAGAGACTGCAGAG-3', *Pak1*-R:5'-GGGGGAACTTCCTGACTAGG-3'). The absence of PAK1 in the PAK1 KO mice was confirmed by Western blot analysis. The mice were housed under a standard 12/12 light/dark cycle condition. All the procedures used for this study were approved by the Animal care committees at the Hospital for Sick Children, Canada and Southeast University, China. AM251, cytochalasin D (Cyto-D), JZL184, Nimesulide, NSC23766 (NSC) and URB597 were from Selleck; D-2-amino-5-phosphonovalerate (D-APV) and Glycyl-H 1152 dihydrochloride (GH1152) were from Tocris; all other drugs (NBQX disodium salt hydrate, (*R*)-(+)-WIN 55,212–2 mesylate salt [WIN], IPA3, TTX, and picrotoxin) were from Sigma.

## Slice electrophysiology

The procedures for the preparation and recording of hippocampal slices were described previously (*Meng et al., 2005*; *Meng et al., 2002*; *Zhou et al., 2011*). Briefly, mouse brains were quickly dissected and transferred to ice-cold artificial cerebrospinal fluid (ACSF) saturated with 95% $O_2$/5% $CO_2$ and sliced to 360 µm sagittal slices. The slices were recovered at 32°C for at least 2 hrs before a single slice was transferred to the recording chamber. ACSF contained (in mM): 120 NaCl, 3.0 KCl, 1.0 $NaH_2PO_4$, 26 $NaHCO_3$, 11 D-glucose, 2.0 $CaCl_2$, and 1.2 $MgSO_4$. To obtain evoked synaptic responses, the stimulation electrode was placed near the stratum pyramidal layer of the CA1 area to execute 0.1Hz stimulation. Whole cell recordings were performed under the voltage clamp mode with a holding potential of −70 mV except in those to construct I/V curves of synaptic currents. Series resistance was monitored by a −3 mV step throughout the entire experiment of whole cell access and if it fluctuated more than 20%, the data were excluded from the analysis. For the E/I ratio experiments, slices were first perfused by ACSF to record total postsynaptic current (PSC) for 10 min, then 10 µM NBQX/50 µM APV were added to specifically record inhibitory postsynaptic current (IPSC), and finally 100 µM picrotoxin was added to verify the inhibitory response. The E/I ratio was calculated as (PSC-IPSC)/IPSC. Spontaneous IPSC (sIPSC) and miniature IPSC(mIPSC) were recorded by including 10 µM NBQX plus 50 µM APV with or without 1 µM TTX in ACSF respectively. Spontaneous EPSC (sEPSC) and miniature EPSC (mEPSC) were recorded with 100 µM picrotoxin with or without 1 µM TTX respectively. For the E/I ratio and s/mIPSC/EPSC (except sIPSC with NSC) experiments, whole-cell recordings were made using glass pipettes (3–5 MΩ) filled with an intracellular solution containing (in mM): 130 CsMeSO₄, 5 NaCl, 1 $MgCl_2$, 0.05 EGTA, 10 HEPES, 1 Mg-ATP, 0.3 $Na_3$-GTP, and 5 QX-314 (pH 7.25) (280–300 mOsm), and for all other measurements, the recordings were made with an intracellular solution containing (in mM): 110 K-gluconate, 25 KCl, 10 $Na_2$-creatine phosphate, 0.2 EGTA, 10 HEPES, 2 Mg-ATP and 0.3 $Na_3$-GTP(pH 7.25) (280–300 mOsm) (*Hashimotodani et al., 2007*; *Huang and Woolley, 2012*). Some of the experiments were repeated in a high Cl intracellular solution containing (in mM): 130 CsCl, 5 NaCl, 10 HEPES, 0.5 EGTA, 5 QX314, 4 Mg-ATP and 0.3 $Na_3$-GTP (pH 7.2–7.4) (280 mOsm) and similar results were obtained, but the data were not included in this study. For whole cell infusion of chemical inhibitors (e.g. peptide, IPA3, and NSC23766), the stock solutions or control vehicle were added to the intracellular solution right before the start of the experiments. The sequence of the PAK1 inhibitory peptides are as follows: active peptide: KKEKERPEISLPSDFEHT; ctrl peptide: GPPARNPRSPVQPPP (final concentration at 20 µg/ml, Genscript). For the DSI experiments, the stimulation frequency was increased to 0.33 Hz and the depolarization was from −70 mV to 0 mV lasting 5 s. For the FAAH inhibitor URB597 experiments, the summary graphs (*Figure 8c,d*) represented pooled data from both females and males (the ratio of females vs males was balanced between WT and PAK1 KO mice). Because the effect of URB597 is more prominent in female rats (*Tabatadze et al., 2015*), we also repeated the URB597 experiment shown in *Figure 8c* in female mice only and the result was the same as the pooled data (data not shown). Synaptic depression in response to sustained high frequency stimulation was induced by 5 Hz lasting 3 min. All data acquisition and analysis were done with pCLAMP and MiniAnalysis programs. All evoked data were normalized to the average of the baseline response.

## Neuronal co-culture and recordings

In order to accurately compare the WT and PAK1 KO neurons, we used a co-culture system where we plated the WT and KO neurons on the same coverslips to keep the culture conditions and other

procedures identical between genotypes. To accomplish this, we crossed *Pak1+/-* mice to *57BL/6-Tg(CAG-EGFP)* to obtain four genotypes: *Pak1+/+/EGFP+*, *Pak1+/+/EGFP-*, *Pak1-/-/EGFP+* and *Pak1-/-/EGFP-*. Hippocampi from *Pak1+/+/EGFP+* and *Pak1-/-/EGFP-* pups or from *Pak1+/+/EGFP-* and *Pak1-/-/EGFP+* were mixed and plated on the same coverslips and the genotype of the neurons was identified based on the presence or absence of EGFP. The procedures for hippocampal culture and recordings were described previously (*Meng et al., 2002*; *Zhou et al., 2011*). Briefly The hippocampi from two pups with suitable genotypes (as described above) from the same litter were dissected and subjected to trypsinization (0.25% at 37°C, 15–20 min), centrifugation (1200 g, 3 min) and resuspension in maintenance medium containing Neurobasal A, 0.5 mM GlutaMax, B27 and 1% penicilin, before being placed on 24-well plate with poly-D-lysine coated glass coverslips. The maintenance medium was half replaced by fresh medium every 4 days. At 12-15DIV coverslips were transferred to a recording chamber containing (in mM): 120 NaCl, 3 KCl, 25 HEPES, 25 Glucose, 1.2 MgCl$_2$, and 2.0 CaCl$_2$ (pH 7.2–7.4) (280 mOsm), and whole cell recordings were made as described above. GABA currents were evoked by 1 mM GABA puff (100 ms) delivered to the cell body through a glass electrode using the pneumatic picopump PV830 (WPI).

## Biochemical assays

Standard methods for extraction and analysis of protein lysates were followed (*Meng et al., 2002*; *Zhou et al., 2011*). Briefly, the brain tissues were dissected quickly in ice-cold 0.1 M PBS and transferred to a homogenizer containing ice-cold RIPA lysis buffer (Beyotime) with 0.5% protease inhibitor cocktail (Roche) and lysed for 45 min at 4°C. Debris was excluded by centrifugation at 15,000 g for 10 min (4°C). For synaptosomal fractions, the protein lysate was first processed by the synaptic protein extraction reagents (Thermo), followed by centrifugations at 1200 g for 10 min to collect the supernatant and additional centrifugations at 15,000 g for 20 min to collect the pellet to be resuspended in RIPA lysis buffer. Proteins were separated on a SDS-PAGE ployacrylamide gel and electrotransfered to a PVDF filter. Filters were then blocked with 5% dry milk TBST (20 mM Tris base, 9% NaCl, 1% Tween-20, pH 7.6) and incubated overnight at 4°C with appropriate primary antibodies in TBST. After washing and incubation with appropriate secondary antibodies, filters were developed using enhanced chemiluminescence (Thermo) method of detection and analyzed using the AlphaEaseFC software as per manufacturer's instruction. Protein loading was further controlled by normalizing each tested protein with actin, α/β-tubulin or GAPDH immunoreactivity on the same blot. Primary antibodies included: anti-PAK1(1:1000, CST, rabbit), anti-GAD2 (1:1000, CST, rabbit), anti-COX-2 (1:3000, CST, rabbit), anti-Actin (1:2000, CST, rabbit), anti-DGLα (1:1000, CST, rabbit) and anti-α/β tubulin (1:3000, CST, rabbit), anti-CB1R (1:1000, Proteintech, rabbit), anti-gephyrin (1:1000; BD, mouse), anti-GAPDH (1:1000; Bioworld, rabbit), anti-MGL (1:1000; Proteintech, rabbit), and anti-FAAH (1:1000; Proteintech, rabbit). n represents the number of independent experiments (i.e. samples from separate mice and tested independently on Western blots).

## Endocannabinoid analysis

Brain regions underwent a lipid extraction process as previously described (*Qi et al., 2015*). In brief, tissue samples were weighed and placed in borosilicate glass culture tubes containing 2 ml of acetonitrile with 5 pmol of [$^2$H$_8$] AEA and 5 nmol of [$^2$H$_8$] 2-AG for extraction. These samples were homogenized with a glass rod, sonicated for 30 min, incubated overnight at -20°C to precipitate proteins, then centrifuged at 1500 g for 5 min to remove particulates. Supernatants were removed to a new glass culture tube and evaporated to dryness under N$_2$ gas, re-suspended in 300 µl of methanol to recapture any lipids adhering to the tube and re-dried again under N$_2$ gas. The final lipid extracts were suspended in 200 µl of methanol and stored at -80°C until analysis. AEA and 2-AG contents within lipid extracts were determined using isotope-dilution, liquid chromatography-tandem mass spectrometry (LC-MS/MS) as previously described (*Qi et al., 2015*). n in the summary graphs of *Figure 7a,b* represents the number of mice.

## Histochemistry and immnostaining of brain sections

The procedure for brain processing and immunohistochemistry were described previously (*Meng et al., 2005*; *Meng et al., 2002*). Briefly, mice were anesthetized by 10% Chloral hydrate, subjected to cardiac perfusion with 0.1 M phosphate-buffered saline (PBS), followed by 4%

paraformaldehyde (PFA, in PBS). The brain was then dissected and transferred to 4% PFA for additional 24 hrs, and then to 30% sucrose solution till it was saturated. The brain was enbeded in Tissue-Tek OCT. compound and frozen by liquid nitrogen before being sliced to 25 μm coronal crystat sections (Leica CM1950). The brain sections were transferred to a glass slide coated with poly-D-lysine for immunostaining. Sections were permeabilized by 0.25% TritonX-100 in 0.1 M PBS (PBT) for 30 min, blocked with 10% fetal bovine serum (FBS) for 1 hr, and incubated with primary antibodies overnight at 4°C, and then appropriate secondary antibodies 2 hrs at 37°C. Primary antibodies used included: anti-GABA (1:200, Sigma, rabbit), anti-VGAT (1:200, CST, rabbit), and anti-gephyrin (1:200, BD, mouse). Cell nucleus was marked with 4, 6-diamidino-2-phenylindole (DAPI, Cayman Chemical). The stained coverslips were mounted using DAKO mounting medium for image collections. Confocal images were obtained on Zeiss LSM 700. For each section, approximately 400 μm width×1200 μm depth of equivalent cortical and hippocampal areas were analyzed to estimate the number of GABA-geric neurons and synapses. For cortical superficial layers, an area of 400 μm width×320 μm depth per section was used. Measurements were performed using Zeiss AimImage Browser software. n represents the number of brain sections and the number of mice for each genotype was no less than 3.

## Immunostaining and image collection/analysis of cultured hippocampal neurons

Hippocampal low-density neuronal cultures were prepared from postnatal day 1 of PAK1 KO and WT littermates as described above (*Meng et al., 2002*; *Zhou et al., 2011*). At 17–18 DIV, culture medium was quickly replaced with ice-cold 4% paraformaldehyde + 4% sucrose for 20 min and permeabilized with 0.25% TritonX-100 for additional 20 min. Cells were then blocked with 3% donkey serum and 3% BSA in PBS for 1 hr, incubated with primary antibodies overnight at 4°C followed by appropriate secondary antibodies for 1 hr at room temperature. After extensive washing with PBS, coverslips were mounted using ProLong Antifade mounting medium (Invitrogen) for image collections. The primary antibodies (1:750 dilution) used for immunostaining were anti-PAK1 (CST, rabbit), anti-COX-2 (CST, rabbit), anti-gephyrin (BD, mouse), anti-PSD-95 (Millipore, mouse) and anti-COX-2 (Santa Cruz, mouse). Secondary antibodies (1:1000 dilution) were: donkey anti-mouse IgG (H+L) Alexa Fluor 488 (Invitrogen) and donkey anti-rabbit IgG (H+L) Alexa Fluor 546 (Invitrogen). Confocal images were obtained on Zeiss LSM 700 at 2048 × 2048 pixels using Zeiss 63× (NA 1.4) objective under the same settings and configurations within each experiment. ImageJ (NIH) software was used for measurements of total fluorescence intensity and puncta staining. 10x30 μm sections of primary dendrites were randomly taken for puncta (with an area of greater than 0.1 μm$^2$) counting. All images were analyzed by experimenters blind to the treatment or genotype of the images. For each experiment, 3 independent cultures from three animals were used for analysis.

## Statistics

All the averaged data were reported as mean ± SEM and statistically evaluated by one-way ANOVA, two-way ANOVA or repeated measures two-way ANOVA, wherever appropriate, followed by post-hoc t-tests. $p < 0.05$ was considered to be significant and indicated with *$p < 0.05$, **$p < 0.01$ or ***$p < 0.001$ in the summary graphs. n represented the number of cells/slices or independent experiments and was used for calculating the degree of freedom. The number of animals was also indicated by the number in the bracket following each n. The statistical data for key summary graphs were provided in the Source data.

## Acknowledgements

This work was supported by grants from the Canadian Institutes of Health Research (CIHR, MOP119421, ZJ and MNH), Canadian Natural Science and Engineering Research Council (NSERC, RGPIN341498, ZJ), the China National Basic Research Program (China 973 Program 2012CB517903, WX and ZJ), Natural Science Foundation of China (NSFC, 31571040, ZZ), and NSFC and CIHR Joint Health Research Initiative Program (81161120543, WX and CCI117959, ZJ). We thank all members of Jia and Xie labs for their technical assistance and comments on the manuscript.

# Additional information

## Competing interests

MNH: A consultant for Pfizer. The other authors declare that no competing interests exist.

## Funding

| Funder | Grant reference number | Author |
|---|---|---|
| National Natural Science Foundation of China | 31571040 | Zikai Zhou |
| Canadian Institutes of Health Research | | Matthew N Hill |
| The China National Basic Research Program | 2012CB517903 | Wei Xie |
| Canadian Institutes of Health Research | MOP119421 | Zhengping Jia |
| Natural Sciences and Engineering Research Council of Canada | RGPIN341498 | Zhengping Jia |

The funders had no role in study design, data collection and interpretation, or the decision to submit the work for publication.

## Author contributions

SX, Acquisition of data, Analysis and interpretation of data, Drafting or revising the article, Performed the experiments, Analyzed the data, Wrote the paper; ZZ, Acquisition of data, Analysis and interpretation of data, Drafting or revising the article, Designed the study, Performed the experiments, Analyzed the data; CL, Acquisition of data, Analysis and interpretation of data, Drafting or revising the article, Performed the experiments, Analyzed the data; YZ, XP, JQ, MM, Acquisition of data, Analysis and interpretation of data, Drafting or revising the article, Performed the experiments; MNH, Acquisition of data, Analysis and interpretation of data, Drafting or revising the article, Wrote the paper; WX, Conception and design, Drafting or revising the article, Designed the study; ZJ, Conception and design, Analysis and interpretation of data, Drafting or revising the article, Designed the study, Wrote the paper

## Author ORCIDs

Zikai Zhou, http://orcid.org/0000-0002-4577-9826
Zhengping Jia, http://orcid.org/0000-0003-4413-5364

## Ethics

Animal experimentation: This study was performed in strict accordance with the recommendations in the Guide for the Care and Use of Laboratory Animals of Canada. All the procedures used for this study were approved by the Animal care committees at the Hospital for Sick Children, Canada and Southeast University, China (protocol #36368).

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
