## [Decision Letter]

Thank you for submitting your work entitled "p21-Activated Kinase 1 Restricts Tonic Endocannabinoid Signaling in the Hippocampus" for consideration by *eLife*. Your article has been reviewed by two peer reviewers, and the evaluation has been overseen by a Reviewing Editor and Richard Aldrich as the Senior Editor.

The reviewers have discussed the reviews with one another and the Reviewing Editor has drafted this decision to help you prepare a revised submission.

The manuscript has been evaluated by two reviewers, who are familiar with the subject of endocannabinoid signaling. Each reviewer provided several positive comments on the study and commented that there are innovative molecular insights into how endocannabinoid signaling is regulated in the hippocampus. The finding that PAK1 limits endocannabinoid signaling through tonic GABAergic transmission is of interest. However, there were several major concerns that were brought up by the reviewers that dampened enthusiasm.

From pharmacological analysis and knockout mice, the main results indicate that inhibition of PAK1 reduces tonic GABergic transmission through anandamide. The second half of the paper focuses on the role of COX-2 as a downstream mediator of PAK1. The reviewers indicated that the data to support a COX-2 mechanism could be strengthened. For example, the western blot data in Figure 7 is not entirely convincing. A major issue is the need for additional support for the localization of COX-2. In particular, there needs to be additional evidence to show COX-2 is working specifically at GABAergic synapses. A related issue is more evidence should be provided for the regulation of COX-2 and how also PAK1 changes COX-2 localization. One suggestion is that PAK1 should be co-localized with gephyrin and PSD-95, as a comparison.

A common concern of both reviewers is the statistical analysis in the manuscript. In some cases, the sample size was deemed low and the methods for analyzing genotype and stimulus intensity require much more clarification and attention. The numbers of cells and animals need to be explicitly cited.

Despite these concerns, we are willing to consider a revision that addresses the issues below more clearly. The referees requested specific demands that were deemed essential. They requested that the revised manuscript demonstrate that:

1) Specificity of COX-2 synaptic action be addressed;

2) The biochemical data showing changes in synaptosomal COX-2 be made more convincing;

3) A more thorough statistical analysis.

*Reviewer #1:*

The work by Xia et al. described a novel mechanism regulating synaptic endocannabinoid signaling at hippocampal CA1 GABAergic synapses. The authors first found that PAK1 deletion or postsynaptic acute inhibition reduced inhibitory transmission via a presynaptic mechanism, thus increasing E/I balance in CA1 hippocampal neurons. Although they found canonical signaling downstream of PAK1, including actin cytoskeletal remodeling, was not involved in the modulation of GABAergic transmission induced by PAK1 deletion, their data implicate endocannabinoid signaling in this process. In support of this novel mechanism, CB1 blockade reveals a larger endocannabinoid tone at CA1 GABAergic synapses after PAK1 deletion or acute postsynaptic pharmacological inhibition. In addition, AEA but not 2-AG levels were increased in PAK1 KO mice. No changes in CB1 receptor function or expression were observed. Based on these data the authors conclude disruption of PAK1 augments AEA tonic signaling to dampen constitutive GABA release. The data to this point are rigorous and compelling and provide a novel molecular mechanism regulating tonic endocannabinoid signaling at CA1 hippocampal GABAergic synapses.

In the second part of the paper, the authors attempt to determine the mechanism by which PAK1 regulates tonic AEA signaling. The authors measure protein levels of MAGL, DAGL, FAAH and COX-2 (but not NAPE-PLD), and based on data that only COX-2 synaptosomal content is reduced, conclude that PAK1 deletion enhances AEA signaling via removal of COX-2 from the synapse. To follow up on this hypothesis, the authors show COX-2 inhibition-induced depression of IPSCs is partially occluded in PAK1 KO mice, and that CB1 blockade reveals similar degree of endocannabinoid tone in the two conditions. Lastly, the authors show that FAAH inhibition, but not MAGL inhibition, also eliminated genotype differences in CB1 antagonist-induced unmasking of endocannabinoid tone, supporting AEA as the tonic signaling regulated by the PAK1-COX-2 signaling system.

Although this second aspect of the work is the most innovative and has the potential to have the highest scientific impact, the execution is relatively weak compared to the first part of the paper. Overall, the evidence presented to support a COX-2 mediated mechanism is far from compelling. Based on this critical issue, as it stands, the data do not support the strong conclusions stated by the authors. Additional studies required to support the conclusions include the following (1 and 2 are critical, 3 may be beyond the scope of the work):

1) The key biochemical data supporting a COX-2 mechanism is based on n=4 sample that synaptosomal COX-2 is decreased in PAK1 KO mice. This is far from convincing. The sample needs to be increased, and full western blots shown. The biochemical data are the key to the paper, since the subsequent electrophysiological data could all be explained if PAK1 was somehow increasing AEA synthesis, rather than affecting COX-2. The argument made against this possibility in the discussion is not convincing, as occlusion of FAAH inhibition effects could occur due to increased synthesis and release of AEA and subsequent activation of the CB1 receptor; the occlusion being measured is at the level of the receptor.

2) Showing in a converging and more convincing way that COX-2 localization is changing when PAK1 is deleted or inhibited. High resolution quantitative imaging in cultured neurons could be used to show this. For example, reduced co-localization between COX-2 and PSD-95 in PAK1 KO mice, or after intracellular dialysis with PAK1 inhibitor.

3) Ideally, determining the mechanisms by which PAK1 affects COX-2 localization. Is this a direct protein-protein interaction, is it phosphorylation? Co-IP experiments and phosphorylation studies should be conducted to test these possibilities.

4) Sample sizes low in many experiments. For example, a key Figure 8 showing MAGL inhibition does not occlude genotype differences in AM251 unmasking of endocannabinoid tome utilizes n=3 for the KO group. This makes the data essentially un-interpretable.

Reviewer #1 (Additional data files and statistical comments):

Statistical analysis are not appropriate in many figures. Sample sizes are also too low in several key experiments.

*Reviewer #2:*

The research paper entitled "p21-activated kinase…signaling in the hippocampus" by Xia et al., demonstrates that PAK1 inhibition suppresses tonic GABAergic transmission via anandamide. The increased tonic anandamide is mediated by suppression of COX-2 at the synapse. These findings are supported by studies using both KO mice and pharmacological experiments. Also, several control experiments are included to strengthen the conclusion. Following are my specific comments.

1) A major question remains to be answered is the specific effect of COX-2 in GABAergic synapses but not in glutamatergic synapses. Authors have not provided a clear explanation for this selective effect. Changes in COX-2 levels cause modification of glutamatergic transmission (Yang et al., Mol. Cell. Neurosci. 37 (2008) 682-695). Also, COX2 is not selective for anandamide (Kozak et al., J. Biol. Chem. 275, 33744-33749, 2000; Kozak et al., J. Biol. Chem 277, 44877-44885, 2002).

2) It is unclear how PAK1 regulates COX-2.

*Reviewer #2 (Additional data files and statistical comments):*

1) Statistics. Time-dependent effects should be analyzed by repeated measures ANOVA. Comparison of one time point using t-test is not adequate. See Figure 1, Figure 6, Figure 8. Also, authors need to present the details of statistics results not just p value.

2) Both number of cells and animals should be provided. It is unclear why n is unbalanced in many experiments. For example, n=10 in wt vs n=16 in KO (Figure 2).

---

## [Author Response]

The manuscript has been evaluated by two reviewers, who are familiar with the subject of endocannabinoid signaling. Each reviewer provided several positive comments on the study and commented that there are innovative molecular insights into how endocannabinoid signaling is regulated in the hippocampus. The finding that PAK1 limits endocannabinoid signaling through tonic GABAergic transmission is of interest. However, there were several major concerns that were brought up by the reviewers that dampened enthusiasm.

From pharmacological analysis and knockout mice, the main results indicate that inhibition of PAK1 reduces tonic GABergic transmission through anandamide. The second half of the paper focuses on the role of COX-2 as a downstream mediator of PAK1. The reviewers indicated that the data to support a COX-2 mechanism could be strengthened. For example, the western blot data in Figure 7 is not entirely convincing. A major issue is the need for additional support for the localization of COX-2. In particular, there needs to be additional evidence to show COX-2 is working specifically at GABAergic synapses. A related issue is more evidence should be provided for the regulation of COX-2 and how also PAK1 changes COX-2 localization. One suggestion is that PAK1 should be co-localized with gephyrin and PSD-95, as a comparison.

A common concern of both reviewers is the statistical analysis in the manuscript. In some cases, the sample size was deemed low and the methods for analyzing genotype and stimulus intensity require much more clarification and attention. The numbers of cells and animals need to be explicitly cited.

Despite these concerns, we are willing to consider a revision that addresses the issues below more clearly. The referees requested specific demands that were deemed essential. They requested that the revised manuscript demonstrate that:

1) Specificity of COX-2 synaptic action be addressed;

As suggested, we have now performed immunostaining experiments in cultured hippocampal neurons to determine the relative distribution of COX-2 and PAK1 at both excitatory and GABAergic synapses and how this was affected by PAK1 disruption. First, we showed that PAK1 was colocalized with PSD95 (excitatory synaptic marker) and (gephyrin (GABAergic synaptic marker) (Figure 9), indicating that PAK1 was localized at both excitatory and inhibitory synapses. In addition, PAK1 was also colocalized with COX-2 (Figure 9). Second, we showed that COX-2 was colocalized with PSD95 (Figure 9), indicating that COX-2 was also expressed at excitatory synapses, but the amount of COX-2 colocalized with PSD95 was not altered in PAK1 KO compared to WT neurons (Figure 9). These results indicate that PAK1 disruption does not affect COX-2 localization at the excitatory synapse. Finally, we showed that COX-2 was also colocalized with gephyrin, and importantly, the amount of this colocalization was significantly reduced in PAK1 KO compared to WT neurons (Figure 9). Taken together, these results suggest that PAK1 regulates COX-2 localization specifically at the inhibitory synapse, which may explain the specific effect of PAK1-COX-2 signaling on inhibitory, but not on excitatory synaptic transmission.

2) The biochemical data showing changes in synaptosomal COX-2 be made more convincing;

As suggested, we have now performed additional experiments to increase the sample size of the synaptosomal COX-2 blots (Figure 7) from 4 to 10. In addition, we have increased the sample sizes of the other proteins analyzed in Figure 7. In support of these biochemical data, we have now performed additional immunostaining experiments using cultured hippocampal neurons and shown that the amount of COX-2 at the inhibitory synapse, but not at the excitatory synapse, was significantly reduced in PAK1 KO neurons (see response to specific demand #1). These results are consistent with the idea that PAK1 regulates COX-2 localization specifically at GABAergic synapses, providing a mechanism for specific effect of PAK1-COX2 signaling on inhibitory synaptic transmission.

3) A more thorough statistical analysis.

As requested, we have performed additional experiments to improve the sample size in the following figures: Figure 1, Figure 1, Figure 3, Figure 3, Figure 3, Figure 4, Figure 5, Figure 5, Figure 6, Figure 7, and Figure 8. We believe that the sample sizes of these experiments are now up to the standard in the field of slice recordings and Western blot analysis. As suggested, we have indicated the number of cells (indicated by n) as well as the number of animals (indicated by the number in the bracket following each n) for each experiment.

As requested, we have performed more thorough statistical analyses, which are detailed below. The statistical data for the relevant figures are provided as source data.

For Figure 1, we used repeated measures two-way ANOVA analysis because we had two factors (genotype and stimulation intensity) and each slice was tested with all stimulation intensities (i.e. repeated measures). Following the ANOVA analysis, we used as post-hoc t-tests to compare WT and KO at various stimuli. Because we were interested in the maximal synaptic responses, the post-hoc t-test for the last stimulation intensity was included in the text.

For Figure 1, Figure 6 and Figure 8, we also used repeated measures two-way ANOVA test. All these experiments were to compare the drug effect on eIPSCs in WT and KO mice. Again we had two factors (genotype and time, or peptide and time). Because each neuron’s response to the drug application was collected over all the time points (10 min time interval was used for statistical analysis), the repeated measures two-way ANOVA analysis was an appropriate test. Following the ANOVA analysis, we then performed post-hoc t-tests between genotypes at various time points, but only the results for the last time point (21-30 interval) were included in the text and in the summary graphs). This time point was chosen because at this time point the drug effect has been stabilized, and therefore the difference between WT and KO at this time point represents the true difference in drug effect between genotypes (rather than differences caused by perfusion speed, penetration of drug into the slice/neurons etc.).

For Figure 3, we used two-way ANOVA analysis followed by post-hoc t-tests. In these experiments we had two independent factors (genotype and drug) and each neuron’s response was measured only once (i.e. with or without drug).

Reviewer #1:

In the second part of the paper, the authors attempt to determine the mechanism by which PAK1 regulates tonic AEA signaling. The authors measure protein levels of MAGL, DAGL, FAAH and COX-2 (but not NAPE-PLD), and based on data that only COX-2 synaptosomal content is reduced, conclude that PAK1 deletion enhances AEA signaling via removal of COX-2 from the synapse. To follow up on this hypothesis, the authors show COX-2 inhibition-induced depression of IPSCs is partially occluded in PAK1 KO mice, and that CB1 blockade reveals similar degree of endocannabinoid tone in the two conditions. Lastly, the authors show that FAAH inhibition, but not MAGL inhibition, also eliminated genotype differences in CB1 antagonist-induced unmasking of endocannabinoid tone, supporting AEA as the tonic signaling regulated by the PAK1-COX-2 signaling system.

Although this second aspect of the work is the most innovative and has the potential to have the highest scientific impact, the execution is relatively weak compared to the first part of the paper. Overall, the evidence presented to support a COX-2 mediated mechanism is far from compelling. Based on this critical issue, as it stands, the data do not support the strong conclusions stated by the authors. Additional studies required to support the conclusions include the following (1 and 2 are critical, 3 may be beyond the scope of the work):

1) The key biochemical data supporting a COX-2 mechanism is based on n=4 sample that synaptosomal COX-2 is decreased in PAK1 KO mice. This is far from convincing. The sample needs to be increased, and full western blots shown. The biochemical data are the key to the paper, since the subsequent electrophysiological data could all be explained if PAK1 was somehow increasing AEA synthesis, rather than affecting COX-2. The argument made against this possibility in the discussion is not convincing, as occlusion of FAAH inhibition effects could occur due to increased synthesis and release of AEA and subsequent activation of the CB1 receptor; the occlusion being measured is at the level of the receptor.

As suggested, we have now performed additional experiments to increase the sample size of the synaptosomal COX-2 blots from 4 to 10 (Figure 7). In addition, we have increased the sample sizes of the other proteins analyzed in Figure 7 (see response to specific demand #2). Full Western blot images for Figure 7 are now provided in Figure 7—figure supplement 1. In support of the Western blot data, we have also performed additional immunostaining experiments using cultured hippocampal neurons and shown that the amount of COX-2 at the inhibitory synapse, but not at the excitatory synapse, was significantly reduced in PAK1 KO neurons (see response to specific demand #1). These results are consistent with the idea that PAK1 regulates COX-2 localization only at GABAergic synapses, providing a mechanism for specific effect of PAK1-COX-2 signaling on inhibitory synaptic transmission.

2) Showing in a converging and more convincing way that COX-2 localization is changing when PAK1 is deleted or inhibited. High resolution quantitative imaging in cultured neurons could be used to show this. For example, reduced co-localization between COX-2 and PSD-95 in PAK1 KO mice, or after intracellular dialysis with PAK1 inhibitor.

As suggested, we have now performed immunostaining experiments using low density cultured hippocampal neurons and the results are described in responses to specific demands #1.

3) Ideally, determining the mechanisms by which PAK1 affects COX-2 localization. Is this a direct protein-protein interaction, is it phosphorylation? Co-IP experiments and phosphorylation studies should be conducted to test these possibilities.

As requested, we have performed immunostaining experiments in low density cultured neurons and the results are described in response to specific demand #1. We have also performed Co-IP experiments using anti-PAK1 and anti-COX-2 antibodies, but unfortunately were not able to show that these two proteins exist in one immunocomplex (data not shown), suggesting that they may not physically interact with each other. Therefore, the effect of PAK1 on COX-2 at the inhibitory synapse is not likely through a protein-protein interaction. We have not been able to analyze the phosphorylation level of COX-2 because we do not have access to phospho-specific antibodies against COX-2.

4) Sample sizes low in many experiments. For example, a key Figure 8 showing MAGL inhibition does not occlude genotype differences in AM251 unmasking of endocannabinoid tome utilizes n=3 for the KO group. This makes the data essentially un-interpretable.

As suggested, we have performed additional experiments to increase the sample sizes of this and other experiments. See response to specific demands #2 and #3.

Reviewer #1 (Additional data files and statistical comments):

Statistical analysis are not appropriate in many figures. Sample sizes are also too low in several key experiments.

See response to specific demand #3.

Reviewer #2:

The research paper entitled "p21-activated kinase…signaling in the hippocampus" by Xia et al., demonstrates that PAK1 inhibition suppresses tonic GABAergic transmission via anandamide. The increased tonic anandamide is mediated by suppression of COX2 at the synapse. These findings are supported by studies using both KO mice and pharmacological experiments. Also, several control experiments are included to strengthen the conclusion. Following are my specific comments.

1) A major question remains to be answered is the specific effect of COX-2 in GABAergic synapses but not in glutamatergic synapses. Authors have not provided a clear explanation for this selective effect. Changes in COX-2 levels cause modification of glutamatergic transmission (Yang et al., Mol. Cell. Neurosci. 37 (2008) 682-695). Also, COX-2 is not selective for anandamide (Kozak et al., J. Biol. Chem. 275, 33744-33749, 2000; Kozak et al., J. Biol. Chem 277, 44877-44885, 2002).

See response to specific demand #1 for explanation of specific effect of PAK1 deletion on GABAergic responses. In brief, although PAK1 and COX-2 are present at both excitatory and inhibitory synapses (Figure 9), the effect of PAK1 deletion on COX-2 expression appears to be specific to COX-2 at the inhibitory synapse (Figure 9). Thus, the level of colocalization of COX-2 with gephyrin, but not with PSD95, was reduced in PAK1 KO compared to WT neurons (Figure 9). Therefore, although COX-2 may function at both inhibitory and excitatory synapses as shown in Yang et al., Mol. Cell. Neurosci. 37 (2008) 682-695, PAK1 deletion leads to reduced COX-2 only at the inhibitory synapse, thus only reduced IPSC, but not EPSC, was observed in PAK1 KO mice. In the present study, the effects of PAK1 deletion and reduced COX-2 expression seem to relate to an exclusive elevation in AEA signalling, but not 2-AG (based on both biochemical analysis of endocannabinoid content and the occlusion of the effects of a FAAH inhibitor, but not a MAGL inhibitor). While the references the reviewer has included do demonstrate that COX-2 has the ability to regulate 2-AG signalling, the work of the Patel and Marnett group examining COX-2 regulation of endocannabinoid signalling (Hermanson et al., 2013 16, 1291-98) has demonstrated that the magnitude of the effect of COX-2 inhibition of AEA signalling is far more robust than what is seen for 2-AG signalling. In fact, the increase in 2-AG content from COX-2 inhibition was only about 5%, while the increase in AEA content was more than double of baseline values (Hermanson et al., 2013). Thus, while we cannot entirely exclude the possibility that the ability of COX-2 activity is capable of producing subtle alterations in 2-AG signalling, our data are consistent with the reports of Hermanson and colleagues in that the impact of reducing COX-2 activity is much more profound for AEA signalling than 2-AG signalling. Alternatively, other pathways targeting COX-2 that are independent of PAK1 may function to regulate 2-AG and these pathways are not altered in PAK1 KO mice. It would be interesting to identify these pathways in the future.

2) It is unclear how PAK1 regulates COX-2.

In both Western blot analysis and immunostaining experiments, the level of COX-2 at the inhibitory synapse, but not the total COX-2 protein, is reduced in PAK1 KO mice. These results suggest that PAK1 regulates COX-2 not likely through affecting its transcription or translation, but rather post-translation mechanisms (such as protein trafficking) specifically at the synapse. The exact mechanism by which PAK1 regulates synaptic COX-2 is unclear, but is not likely through a direct protein-protein interaction because we were not able to show that PAK1 and COX-2 exist in one immunocomplex in the Co-IP experiments.

Reviewer #2 (Additional data files and statistical comments):

1) Statistics. Time-dependent effects should be analyzed by repeated measures ANOVA. Comparison of one time point using t-test is not adequate. See Figure 1, Figure 6, Figure 8. Also, authors need to present the details of statistics results not just p value.

As suggested, we have performed more statistical analyses. See response to specific demand #3 for details.

*2) Both number of cells and animals should be provided. It is unclear why n is unbalanced in many experiments. For example, n=10 in wt vs n=16 in KO (Figure 2).*

As requested, both the number of cells or slices (indicated by “n”) and number of animals (number in the bracket following each “n”) are provided in the figure legend. In some experiments, the n number of WT and KO was unbalanced because of the availability of the mice, sometimes due to an unbalanced number of WT and KO littermates from the *PAK1+/-* breeders.